# Homolytic H$_2$ dissociation for enhanced hydrogenation catalysis on oxides

Chengsheng Yang[1], Sicong Ma [2] ✉, Yongmei Liu[1], Lihua Wang[3], Desheng Yuan[1], Wei-Peng Shao [4], Lunjia Zhang[4], Fan Yang [4], Tiejun Lin [5], Hongxin Ding [1], Heyong He[1], Zhi-Pan Liu [1,2], Yong Cao[1], Yifeng Zhu [1] ✉ & Xinhe Bao [1,6] ✉

The limited surface coverage and activity of active hydrides on oxide surfaces pose challenges for efficient hydrogenation reactions. Herein, we quantitatively distinguish the long-puzzling homolytic dissociation of hydrogen from the heterolytic pathway on Ga$_2$O$_3$, that is useful for enhancing hydrogenation ability of oxides. By combining transient kinetic analysis with infrared and mass spectroscopies, we identify the catalytic role of coordinatively unsaturated Ga$^{3+}$ in homolytic H$_2$ dissociation, which is formed in-situ during the initial heterolytic dissociation. This site facilitates easy hydrogen dissociation at low temperatures, resulting in a high hydride coverage on Ga$_2$O$_3$ (H/surface Ga$^{3+}$ ratio of 1.6 and H/OH ratio of 5.6). The effectiveness of homolytic dissociation is governed by the Ga-Ga distance, which is strongly influenced by the initial coordination of Ga$^{3+}$. Consequently, by tuning the coordination of active Ga$^{3+}$ species as well as the coverage and activity of hydrides, we achieve enhanced hydrogenation of CO$_2$ to CO, methanol or light olefins by 4-6 times.

Hydrogen activation on solid surfaces is a central topic in catalytic processes involving hydrogen evolution, such as hydrogenation[1–3] and hydrogen production[4,5], and for fundamental mechanistic studies to develop catalyst design principles[6,7].

Many studies have demonstrated that hydrogen is easily homolytically dissociated into hydrides on a single metal atom or ensembles with several metal atoms for hydrogenation reactions. However, while these supported metals exhibit high activity, they often have low selectivity due to over-hydrogenation[8]. Recently, oxides have emerged as a promising class of catalysts for hydrogen-related reactions including CO$_2$ hydrogenation[3], syngas conversion[9–11], and propane dehydrogenation[12], exhibiting both remarkable activity and selectivity. Metal cations on oxides are strictly isolated by the counter oxygen anions, resulting in the predominantly considered heterolytic

dissociation mechanism of hydrogen[13]. The infrared spectroscopy (IR)[14,15], inelastic neutron scattering[16,17], and solid-state NMR[18] have evidenced the presence of heterolytic dissociation of H$_2$ on oxides including In$_2$O$_3$[19], CeO$_2$[20,21], MgO[15], ZnO[14], and Ga$_2$O$_3$[18,22,23].

However, the heterolytic dissociation presents a limit for the subsequent hydrogenation on oxides, as the amount of resulting hydride may not be sufficient for the reactions involving transfer of multiple electrons and hydrogen atoms. The migration of hydrides from proximal sites[24–26], such as spillover, ensures the catalytic cycle, but would result in high barriers up to 2-8 eV and slow down the reaction[27,28]. In addition, the migration of hydrides along lattice oxygen atoms increases the possibility of hydride elimination to hydroxyl, further exacerbating the hydride shortage[29]. To the present, rare definitive experimental proof has quantified the evolution of hydrogen

[1]Department of Chemistry, Shanghai Key Laboratory of Molecular Catalysis and Innovative Materials, Collaborative Innovation Center of Chemistry for Energy Materials, Fudan University, Shanghai 200438, China. [2]Key Laboratory of Synthetic and Self-Assembly Chemistry for Organic Functional Molecules, Shanghai Institute of Organic Chemistry, Chinese Academy of Sciences, Shanghai 200032, China. [3]Shanghai Synchrotron Radiation Facility, Shanghai Advanced Research Institute, Chinese Academy of Sciences, Shanghai 201204, China. [4]School of Physical Science and Technology, Shanghai Tech University, Shanghai 201210, China. [5]Key Laboratory of Low-Carbon Conversion Science and Engineering, Shanghai Advanced Research Institute, Chinese Academy of Sciences, Shanghai 201210, China. [6]State Key Laboratory of Catalysis, National Laboratory for Clean Energy, Collaborative Innovation Center of Chemistry for Energy Materials, Dalian Institute of Chemical Physics, Chinese Academy of Sciences, Dalian 116023, China. ✉e-mail: scma@mail.sioc.ac.cn; zhuyifeng@fudan.edu.cn; xhbao@dicp.ac.cn

on oxides, perpetuating the dilemma of hydrogen dissociation mechanism. The quantitative and time-resolved analyses of the complex dynamics of hydrogen activation remains a crucial challenge in understanding the hydrogen activation and increasing the hydrogenation ability of oxides.

Herein, we developed a spectroscopic kinetic approach that combines transient kinetic analysis with infrared spectroscopy (TKA-IR) and mass spectroscopy (TKA-MS), to differentiate the long-puzzling homolytic dissociation of hydrogen from the heterolytic pathway on $Ga_2O_3$. We showed that the homolytic dissociation of hydrogen emerges on the coordinatively unsaturated $Ga^{3+}$ that formed in-situ during the initial heterolytic dissociation process. The homolytic dissociation sites are highly dependent on the initial coordination of metal cations. The octahedral $Ga^{3+}$ ($Ga_{[Oct]}$) on $\alpha$-$Ga_2O_3$ gives a high H/surface $Ga^{3+}$ ratio of 1.6 and H/OH ratio of 5.6 as a result of the homolytic dissociation, improving the hydrogenation ability for oxides. We were therefore able to achieve greatly enhanced hydrogenation catalysis by tuning the coordination of active $Ga^{3+}$, and the coverage and activity of activated hydrides.

## Results

### Surface coordination structures of $Ga_2O_3$

$Ga_2O_3$ is a well applied material for hydrogen evolution related catalysis[30,31]. We synthesized $Ga_2O_3$ nanoparticles with different crystalline structures ($\alpha$-, $\varepsilon$-, and $\beta$-$Ga_2O_3$). The identification of crystal phases was confirmed by X-ray diffraction (XRD, Supplementary Fig. 1) and high-resolution transmission electron microscopy (HRTEM, Supplementary Fig. 2) with selected area electron diffractions. The sizes of $\alpha$-$Ga_2O_3$ and $\varepsilon$-$Ga_2O_3$ crystallites were determined to be 9-10 nm based on the Scherrer equation, while $\beta$-$Ga_2O_3$ had a mean size of 45 nm. HRTEM revealed that $\alpha$-$Ga_2O_3$ was mesoporous nanorods composing of ~10 nm nanoparticles, $\varepsilon$-$Ga_2O_3$ was aggregates composing of ~15 nm

nanoparticles, and $\beta$-$Ga_2O_3$ displayed irregular shapes. Consequently, the $Ga_2O_3$ catalysts exhibited varying surface areas within the range of 47-95 $m^2/g$ (Supplementary Table 1). To characterize the coordination structures of $Ga^{3+}$ cations in $Ga_2O_3$, X-ray absorption near-edge structure (XANES) was employed (Supplementary Figs. 3-4, Supplementary Table 2). The absorptions at around 10375 and 10379 eV are attributed to tetrahedral $Ga^{3+}$ ($Ga_{[Tet]}$) and octahedral $Ga_{[Oct]}$, respectively[32]. The results showed that $\alpha$-$Ga_2O_3$ is dominated by octahedral coordination, $\varepsilon$-$Ga_2O_3$ has a proportion of octahedral sites of 85%, whereas $\beta$-$Ga_2O_3$ has 50% octahedral and 50% tetrahedral sites. The results were consistent with the structures of ideal crystals.

However, the surface coordination of $Ga^{3+}$ sites may vary due to structural defects or perturbations of the adsorbed species[33]. To investigate the surface coordination, we conducted hydrogen-adsorbed infrared spectroscopy ($H_2$-IR). We degassed the catalysts in the Ar flow at 350 °C and then exposed them to $H_2$ for adsorption. The $H_2$-IR spectra revealed two IR bands at 2003 and 1980 $cm^{-1}$, which were assigned to hydrides bonding to $Ga_{[Tet]}$ and $Ga_{[Oct]}$, respectively (Fig. 1a–c)[22]. The assignment was validated using deuterium-adsorbed IR ($D_2$-IR) operating under the same conditions, which gave the shifted $Ga_{[Tet]}$-D and $Ga_{[Oct]}$-D bands at 1434 and 1420 $cm^{-1}$ (Supplementary Fig. 5)[34]. The $H_2$-$D_2$ exchange monitored by IR showed the interconversion of the Ga-H and Ga-D bands upon gas switching, further verifying the assignments of Ga-H species (Supplementary Fig. 6). We also observed the appearance of O-H and O-D groups linked to tetrahedral and octahedral $Ga^{3+}$ upon $H_2$ and $D_2$ adsorption, as evidenced by bands at 3656 and 3736 $cm^{-1}$, and at 2689 and 2781 $cm^{-1}$, respectively (Supplementary Figs. 7–9)[35]. By employing quantitative fitting of the IR spectra, we determined that the proportions of octahedral $Ga^{3+}$ on the surfaces are 88%, 46%, and 17% for $\alpha$-, $\varepsilon$-, and $\beta$-$Ga_2O_3$, respectively (Supplementary Table 2).

The prevalence of octahedral sites on $Ga_2O_3$ significantly influences the evolution of surface structure and formation of oxygen

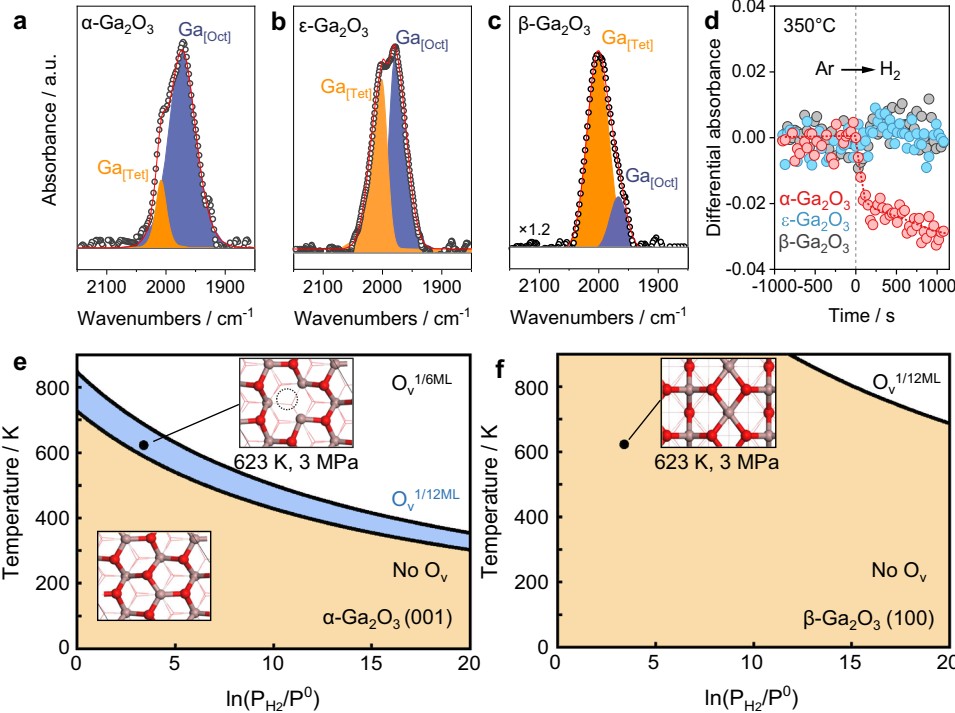

**Fig. 1 | Surface structural characterizations. a–c** $H_2$-IR for $Ga_2O_3$ samples contacting with $H_2$ at 350 °C. **d** Differential absorbance of white line peak in Ga K-edge XANES for $Ga_2O_3$ samples when changing atmosphere from Ar to $H_2$ at 350 °C. The difference spectrum is collected at time delay 30 s after subtracting Ar-treatment spectra. Thermodynamic phase diagram for $\alpha$-$Ga_2O_3$ (001) (**e**) and $\beta$-$Ga_2O_3$ (100) (**f**)

contacting with $H_2$ at different temperatures and $H_2$ partial pressures ($P_{H_2}$), the phase diagram is computed based on Gibbs free energy data for the reaction ($Ga_2O_3$ + $xH_2$ → $Ga_2O_{3-x}$ + $xH_2O$) from DFT calculations assuming a $H_2O$ pressure of 0.1 kPa. $P^0$ represents the standard pressure (101.325 kPa). Source data are provided as a Source Data file.

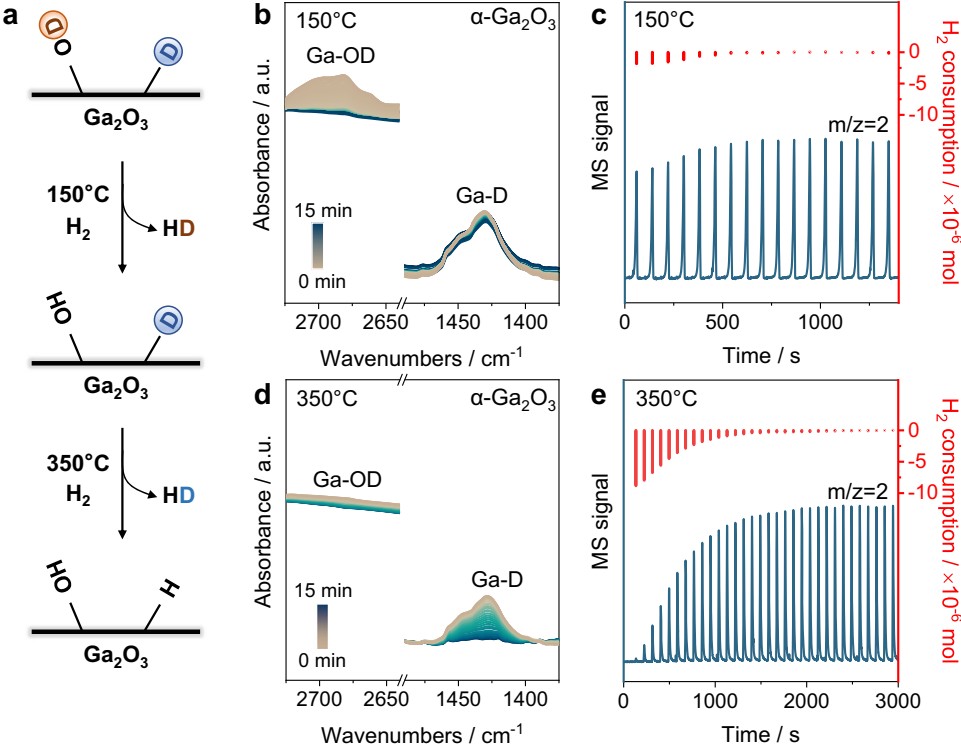

**Fig. 2 | Quantitative analysis of hydride and hydroxyl species on Ga₂O₃ surface.**
**a** The scheme illustrating temperature-dependent exchange chemistry of surface deuterides and deuterated hydroxyl groups with gaseous H₂. **b** IR spectra of α-Ga₂O₃ during H₂-exchange at 150 °C after the catalysts were saturated with D₂ at 350 °C. **c** TKA-MS (Blue) and the amount of H₂ consumption (Red) during H₂-exchange at 150 °C after α-Ga₂O₃ were saturated with D₂ at 350 °C. **d** IR spectra of α-Ga₂O₃ during H₂-exchange at 350 °C after the catalysts were saturated with D₂ at 350 °C. **e** TKA-MS (Blue) and the amount of H₂ consumption (Red) during H₂-exchange at 350 °C after α-Ga₂O₃ were saturated with D₂ at 350 °C. Source data are provided as a Source Data file.

vacancy ($O_v$), as evidenced by in-situ X-ray photoelectron spectroscopy (XPS) and time-resolved energy dispersive X-ray absorption spectroscopy (EDXAS). The catalyst was treated under 3 MPa H₂ at 350 °C and was carefully transferred to the chamber without exposing to air for XPS measurements. Our results show that the H₂ treatment led to a decrease of surface O to Ga ratio (O/Ga) on α-Ga₂O₃ from 1.43 to 1.18, corresponding to $O_v$ concentration of 17.5% (Supplementary Fig. 10)[36]. In contrast, we observed fewer changes in O/Ga for ε-Ga₂O₃ ($O_v$ concentration of 8.2%) and β-Ga₂O₃ (trace amounts of $O_v$) compared to α-Ga₂O₃, confirming a greater preference for the formation of oxygen vacancies and undercoordinated Ga³⁺ on α-Ga₂O₃ (Supplementary Fig. S10). The energy dispersive mode of EDXAS enables the tracking of fast structural change on catalysts induced by hydrogen with second-level resolution. The differential treatment of Ga K-edge XANES spectra showed a rapid decrease in white line intensity upon H₂ treatment for α-Ga₂O₃ (Fig. 1d and Supplementary Fig. 4), supporting the formation of $O_v$ observed in XPS. The temperature-programmed reduction (TPR) also revealed a higher reduction degree for α-Ga₂O₃ (Supplementary Fig. 11).

Ab-initio thermodynamics analyses were performed on the most stable α-Ga₂O₃ (001), ε-Ga₂O₃ (011) and β-Ga₂O₃ (100) surfaces (Supplementary Data 1, Supplementary Table 3 and Supplementary Fig. 12) to determine the Gibbs free energy change ($\Delta G$) for the formation of oxygen vacancies as the function of temperature and H₂ partial pressure (Fig. 1e, f). α-Ga₂O₃ (001) features honeycomb-interlinked Ga-O-Ga six-membered rings exposing Ga [Oct] and three-coordinated $O_{3c}$ atoms in surface. The ε-Ga₂O₃ (011) exposes the 50% four- and 50% five-coordinated Ga[Oct] atoms. And β-Ga₂O₃ (100) shows the penta-membered ring interlinked pattern with the exposure of mixed pattern of Ga [Oct] and Ga [Tet] accompanied with $O_{3c}$ atoms in surface (Supplementary Figs. 13–14). Under typical catalytic hydrogenation

conditions of 350 °C and 3 MPa H₂, α-Ga₂O₃ (001) surface is more prone to loss surface $O_{3c}$ atom with the surface $O_v$ coverage of 1/12-1/6 monolayer. In contrast, the β-Ga₂O₃ (100) is less likely to lose oxygen atoms at the same conditions, as supported by XPS and XANES data. This difference in surface Ga³⁺ coordination environment leads to distinct pathways for H₂ activation.

## Homolytic H₂ dissociation initiated by in-situ formed under-coordinated Ga³⁺

The presence of hydride and hydroxyl species has been detected through H₂-IR during the exposure of Ga₂O₃ to H₂, indicating a potentially heterolytic pathway for H₂ activation (Supplementary Figs. 7–9). However, our quantitative analysis of surface hydride and hydroxyl concentrations revealed inconsistencies. The quantification was made based on the temperature-dependent exchange chemistry of surface deuterides and deuterated hydroxyls with gaseous H₂ (Fig. 2a and Supplementary Fig. 15). During the H₂-D₂ exchange experiment, we noticed that Ga-OD at ca. 2670 cm⁻¹ could be fully exchanged at 150 °C (Ga-OD + H₂ → Ga-OH + HD) while Ga-D at ca. 1410 cm⁻¹ remained intact at this temperature (Fig. 2b and Supplementary Fig. 16a). Upon elevating the temperature to 350 °C, the remaining Ga-D was fully exchanged with H₂ forming HD with *m/z* of 3 (Ga-D + H₂ → Ga-H + HD, Fig. 2d and Supplementary Fig. 16b). The corresponding H₂ consumptions were then measured for the quantification using TKA-MS operating at 150 and 350 °C (Fig. 2c, e and Supplementary Fig. 17, see Supplementary note 1 for details). The results reveal that the hydrogen dissociative adsorption capacity of α-Ga₂O₃ is 20 times higher than that of β-Ga₂O₃ (Supplementary Table 1). Moreover, the amounts of hydride and hydroxyl on α-Ga₂O₃ were 1.50 and 0.27 mmol/g respectively, with the hydride-to-hydroxyl ratio deviating greatly from the expected stochiometric value of 1 in

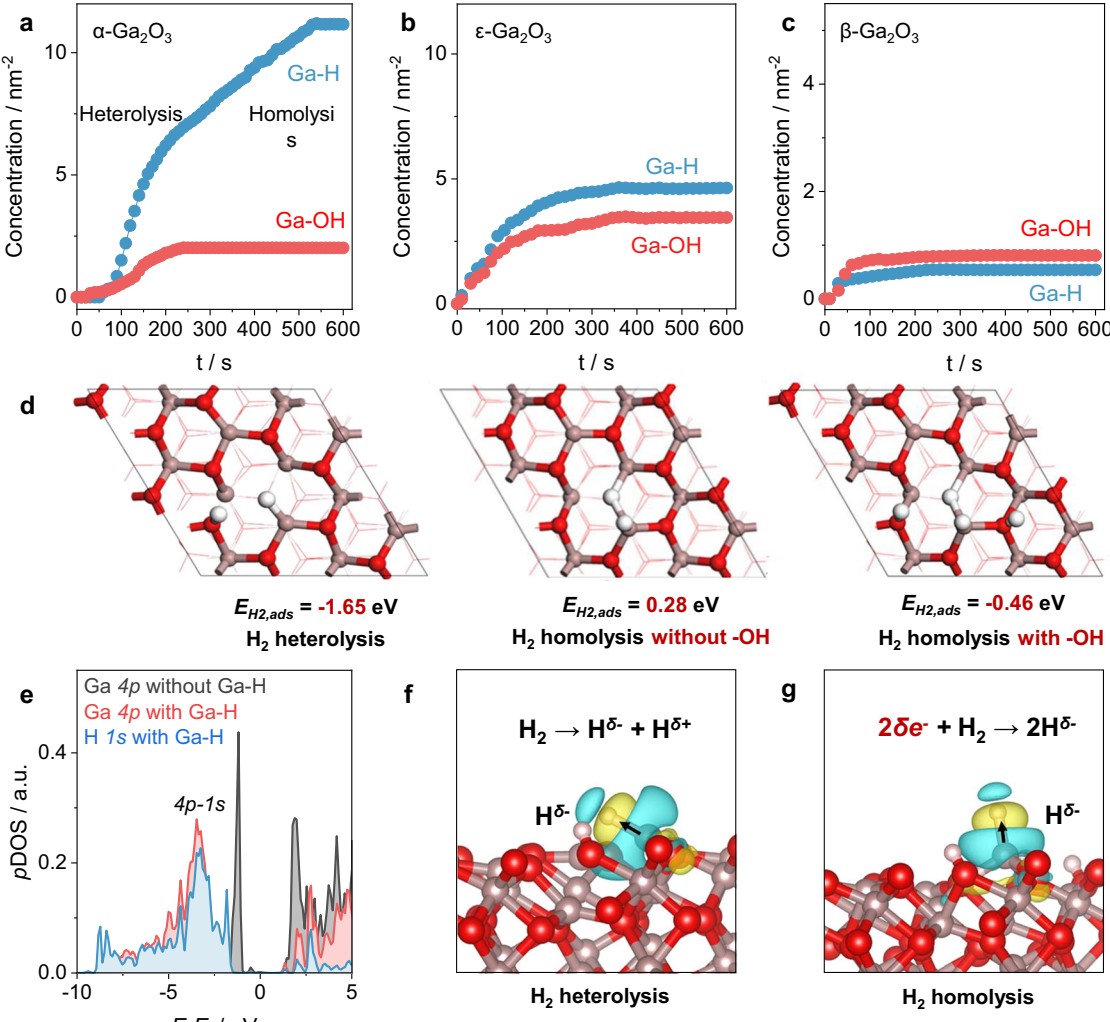

**Fig. 3 | Dynamics of H₂ dissociation for Ga₂O₃. a–c** The evolution of surface Ga-H and Ga-OH over Ga₂O₃ samples with contact time of H₂ at 350 ˚C. **d** H₂ adsorption energy by homolytic dissociation over α-Ga₂O₃ (001). Red ball: O atom, gray ball: Ga atom, white ball: H atom. **e** Projected density of states of the Ga *4p* orbital and H *1s* orbital before and after the formation of GaHₓ hydride species on O-defective α- Ga₂O₃ (001). Charge density difference contour plots for heterolytic (**f**) and homolytic dissociation (**g**) of H₂ on O-defective α-Ga₂O₃ (001). The cyan and yellow colors indicate the decrease and increase in the electron density, respectively. The 3D iso-surface value is set as 0.0014 e Å⁻³. Source data are provided as a Source Data file.

heterolytic dissociation. Remarkably, the ratio of hydride to surface Ga³⁺ for α-Ga₂O₃ is as high as 1.6, while this ratio of ε-, β-Ga₂O₃ is 0.67 and 0.08 respectively, suggesting the formation of high coverage GaHₓ complexes (x > 1) over α-Ga₂O₃ (Supplementary Tables 4–7). These unexpected findings led us to speculate the presence of a second process producing extra hydrides on α-Ga₂O₃, i.e., the homolytic dissociation of hydrogen.

We developed a fast time-response cell for time-resolved TKA-IR, allowing us to track the dynamic evolution of H₂ dissociation and understand the origin of GaHₓ complexes. This approach provided compelling evidences confirming the involvement of the homolytic dissociation pathway. The activated Ga₂O₃ catalysts were first flushed with Ar flow at 350 ˚C until stable and then replaced by H₂ or D₂ flow with an Ar residual time of ~20 s (Supplementary Fig. 18). As shown in Fig. 3a–c, the red plots correspond to the evolution of concentrations of hydroxyl species and the blue plots are those for hydride species. Upon exposure of Ga₂O₃ to H₂, both Ga-H and Ga-OH form rapidly and intensify concurrently. In the meantime, adsorbed water was detected within the initial 10 s for all 3 catalysts, suggesting the formation of oxygen vacancies and undercoordinated Ga³⁺ at this stage (Supplementary Fig. 19). For α-Ga₂O₃, Ga-H concentration

increased at a rate of 3.0 nm⁻²·min⁻¹, while Ga-OH developed at an average rate of 0.5 nm⁻²·min⁻¹, reaching a plateau at 240 s. However, the concentration of Ga-H species continued to rise at a rate of 0.9 nm⁻²·min⁻¹ even after Ga-OH species reached stability. Ga-H species of α-Ga₂O₃ finally reached a plateau at 530 s, giving a high hydride-to-hydroxyl ratio of 5.6. In contrast, the saturation of ε-Ga₂O₃ surface occurred around 200–250 s, exhibiting a similar evolution to Ga-OH on α-Ga₂O₃. The final hydride-to-hydroxyl ratio was 1.3, slightly higher than the stochiometric ratio of 1 for heterolytic dissociation. For the contrasting case of β-Ga₂O₃, both Ga-H and Ga-OH species generated at an average rate of 0.7 nm⁻²·min⁻¹. The surface of β-Ga₂O₃ reached saturation after 50 s with a hydride-to-hydroxyl ratio of 0.7. The quantitative and time-dependent results of hydrides and hydroxyls clearly showed that α-Ga₂O₃ opens a homolytic pathway for H₂ dissociation. These results contrast with the general preference for heterolytic dissociation of H₂ on oxides[37].

The density functional theory (DFT) calculation is further performed to understand H₂ dissociation on O-defective α-Ga₂O₃ (001). As illustrated in Fig. 3d, the H₂ heterolytic dissociation releases large energies of 1.65 eV first. The resulting hydride tends to migrate to the neighboring O atom, forming hydroxyl and releasing energy of

0.58 eV. Interestingly, we found that $H_2$ homolytic dissociation in the presence of neighboring hydroxyl groups with the final Ga-H/OH ratio of 1:1 shows a negative dissociation energy of -0.46 eV (Fig. 3d). Further increasing the Ga-H/OH ratio to 2:1 also shows the negative dissociation energy (-0.55 eV, Supplementary Fig. 20). On the contrary, $H_2$ homolytic dissociation without the prior $H_2$ heterolytic dissociation shows the positive dissociation energy (+0.28 eV). It suggests that $H_2$ heterolytic dissociation is the perquisite for homolytic dissociation (Supplementary Fig. 21).

To understand the promotion effect of hydroxyls in stabilizing Ga-H, we analyzed the projected density of states of the $Ga^{3+}$ before and after homolytic $H_2$ dissociation (Fig. 3e). The presence of hydroxyl and $O_v$ leads to the large electron occupation of the valence band maximum on Ga $4p$ orbital with the Bader charge of 2.21 $e^-$. Once homolytic dissociation occurs, a covalent bond between Ga $4p$ and H $1s$ orbital is established and the electrons transfer from the high-energy-level Ga $4p$ orbital to the low-energy-level $4p$-$1s$ hybrid orbital, leading the decrease of Bader charge to 1.66 $e^-$ for Ga atom on $4p$ orbital. Moreover, the charge density difference contour plots confirm the significant local electron transfer from Ga to H atom upon formation of Ga-H species (Fig. 3f–g). In the case of homolytic dissociation, the presence of hydroxyl and $O_v$, facilitates the transfer of more electrons from Ga to H through the reaction of $2\delta e^- + H_2 \rightarrow 2H^{\delta-}$ (where $\delta$ is around 0.25-0.27). The extra electrons, concurrent with the formation of neighboring hydroxyl and $O_v$, stabilize the Ga-H species and promote homolytic dissociation. Homolytic dissociation of $H_2$ is also thermodynamically favored on O-defective $\epsilon$-$Ga_2O_3$ (011) surface with the $H_2$ dissociation energy of −0.55 eV and the charge transfer from Ga to H (Supplementary Figs. 22−23 and Supplementary Table 8), whereas the $\beta$-$Ga_2O_3$ (100) surface cannot dissociate the $H_2$.

Consequently, a high coverage of hydride becomes thermodynamically favorable, which is essential for producing active hydrides for hydrogenation reactions. The time-resolved IR shows the Ga-H peak shifted towards higher wavenumbers of ca. 12 $cm^{-1}$ during the exposure of $\alpha$-$Ga_2O_3$ to $H_2$, whereas only slight offsets were observed for $\epsilon$-$Ga_2O_3$ and $\beta$-$Ga_2O_3$ (Supplementary Figs. 24−25). The Ga-H band shifts supported the increase of hydride coverage as the progress of homolytic dissociation. It is worth noting that the coverage of H* on metal ions was calculated close to saturation. When considering the coverage of H* on surface O of oxide (Supplementary Table 7), there still allows potential in promoting the adsorption capacity of H*.

We further investigated the preference for homolytic $H_2$ dissociation on octahedral $Ga^{3+}$ sites. The homolytic $H_2$ dissociation on different $Ga_2O_3$ crystalline phases ($\alpha$ and $\beta$) was checked. Upon the formation of $O_v$ on $\alpha$-$Ga_2O_3$ (001), the Ga atoms are at a distance of 2.89 Å providing a suitable environment for homolytic $H_2$ dissociation (Supplementary Fig. 26). Differing from the O-defective $\alpha$-$Ga_2O_3$ (001) surface, the Ga-Ga distances on other surfaces are too large (e.g., 3.08 Å on $\beta$-$Ga_2O_3$ (100) surface) to dissociate $H_2$ (Supplementary Fig. 26).

## Promotion effect of $H_2$ homolytic dissociation in $CO_2$ hydrogenation

The homolytic dissociation of $H_2$ leads to the formation of highly active hydride species. As evidenced by in-situ XPS in Fig. 4a, $H_2$ treatment on $\alpha$-$Ga_2O_3$ gave a shoulder at the binding energy 2.5 eV higher than the central peak in Ga $3d$ spectrum. This is attributed to the presence of Ga-H species with the electron transfer from Ga $4p$ to H $1s$, where the decrease in the number of outer-shell electrons of Ga enhances the attraction of the nucleus to inner-shell electrons. Indeed, the core-level chemical shift of simulated XPS based on DFT calculation shows that the existence of Ga-H species leads to a core-level chemical shift of 0.56 eV in the higher binding energy region, which is far greater than the shifts when $O_v$ or hydroxyl groups were present

(Supplementary Table 9). Similar phenomena have been observed on $CeO_2$[21,38,39]. The obvious shoulder peak on $\alpha$-$Ga_2O_3$ suggests the massive formation of Ga-H bond, while the feature of Ga-H species is less pronounced for $\epsilon$-$Ga_2O_3$ and negligible for $\beta$-$Ga_2O_3$, due to the lower coverages of hydride in these two cases. $H_2$ chemisorption isotherms obtained by IR (Fig. 4b and Supplementary Fig. 27) show that the formation of Ga-H for $\alpha$-$Ga_2O_3$ occurs at $H_2$ pressure as low as 0.086 mbar. Conversely, the formation of Ga-H for $\epsilon$-$Ga_2O_3$ and $\beta$-$Ga_2O_3$ does not happen even at elevated pressure up to 10 mbar. During temperature-programmed surface reaction of $H_2$ ($H_2$-TPSR, Fig. 4c), the exchange of Ga-D and $H_2$ on $\alpha$-$Ga_2O_3$ begins at ~50 °C while that on $\beta$-$Ga_2O_3$ initiates at ~250 °C, implying the high reactivity of Ga-D with hydrogen on $\alpha$-$Ga_2O_3$. These results suggest hydrogen easily undergoes dissociative adsorption on $\alpha$-$Ga_2O_3$. The resulting hydrides formed by homolytic dissociation have higher surface coverage and reactivity than those from heterolytic dissociation, thereby potentially enhancing the rate of hydrogenation reactions.

To verify our hypothesis regarding the promotion effect of active hydrides in hydrogenation reactions, representative $CO_2$ hydrogenations were studied, including the reverse water gas shift reaction (rWGS), methanol synthesis and the recently discovered oxide-zeolite-based $CO_2$-to-light olefins (OX-ZEO). In the rWGS reaction, CO formation rate on $\alpha$-$Ga_2O_3$ was 10.5 $mol \cdot mol_{Ga}^{-1} \cdot h^{-1}$. In comparison, $\epsilon$-$Ga_2O_3$ and $\beta$-$Ga_2O_3$ exhibited CO formation rates of 5.7 $mol \cdot mol_{Ga}^{-1} \cdot h^{-1}$ and 1.5 $mol \cdot mol_{Ga}^{-1} \cdot h^{-1}$, respectively. In methanol synthesis, $\alpha$-$Ga_2O_3$ showed a methanol formation rate of 0.62 $mol \cdot mol_{Ga}^{-1} \cdot h^{-1}$, while $\beta$-$Ga_2O_3$ had almost no activity in synthesizing methanol. Both in low-pressure rWGS and high-pressure methanol synthesis, $\alpha$-$Ga_2O_3$ showed more than 6 times higher CO or methanol formation rate per $Ga^{3+}$ site than $\beta$-$Ga_2O_3$ at 350 °C (Fig. 4d, e and Supplementary Fig. 28).

In OX-ZEO catalysis, $Ga_2O_3$ was physically mixed with SAPO-18 zeolite to selectively convert $CO_2$ into light olefins ($C_2^= $-$C_4^=$) (Fig. 4f and Supplementary Fig. 29). Light olefins are expected via $CO_2$ hydrogenation on oxide and subsequent C-C coupling of intermediates in zeolites. The selectivity to light olefins for $\alpha$-$Ga_2O_3$ is 26.2% in the total products and 80.1% in the hydrocarbons at the $CO_2$ conversion of 12.5%. The light olefins selectivity for $\epsilon$-$Ga_2O_3$ is 11.2% in the total products and 68.8% in the hydrocarbons at a $CO_2$ conversion of 9.3%. The $\beta$-$Ga_2O_3$ gave a low olefin selectivity of 2.9% in the total products and 58.3% in the hydrocarbons at a conversion of 7.7%. The formation rates of light olefins are further measured at conversions lower than 5%. It was shown that, in terms of olefin formation rates, $\alpha$-$Ga_2O_3$ was 0.6 times higher than $\epsilon$-$Ga_2O_3$ and 3.8 times higher than $\beta$-$Ga_2O_3$. These findings indicate that $GaH_x$ derived from homolytic dissociation is efficient in hydrogenating $CO_2$ into intermediates, which is in consistent with the observations in rWGS and methanol synthesis.

The plot of light olefins selectivity as a function of $CO_2$ conversion provides further evidence of the higher hydrogenation ability of active $GaH_x$ over $\alpha$-$Ga_2O_3$ (Supplementary Fig. 30). For $\alpha$-$Ga_2O_3$, the selectivity of light olefins decreased with the increase of $CO_2$ conversion, which is due to the over-hydrogenation of olefins to paraffins. In contrast, the olefin selectivity for $\beta$-$Ga_2O_3$ was at a low level and increased slightly with the elevation of $CO_2$ conversion (Supplementary Figs. 31−33). Moreover, the selectivity of paraffins did not show a significant increase with increasing $CO_2$ conversion, indicating the intrinsic low activity of $\beta$-$Ga_2O_3$ in hydrogen activation (Supplementary Fig. 34).

The high reactivity of active $GaH_x$ over $\alpha$-$Ga_2O_3$ was also demonstrated with CO hydrogenation to light olefins. The selectivity to light olefins for $\alpha$-$Ga_2O_3$ was 63.4% in the hydrocarbons at the CO conversion of 10.6% (Supplementary Fig. 35a). $\beta$-$Ga_2O_3$ gave a low CO conversion of 1.4% and selectivity to light olefins of 50.4%. The formation rate of light olefins for $\alpha$-$Ga_2O_3$ was 6.3 times higher than $\beta$-$Ga_2O_3$ (Supplementary Fig. 35b).

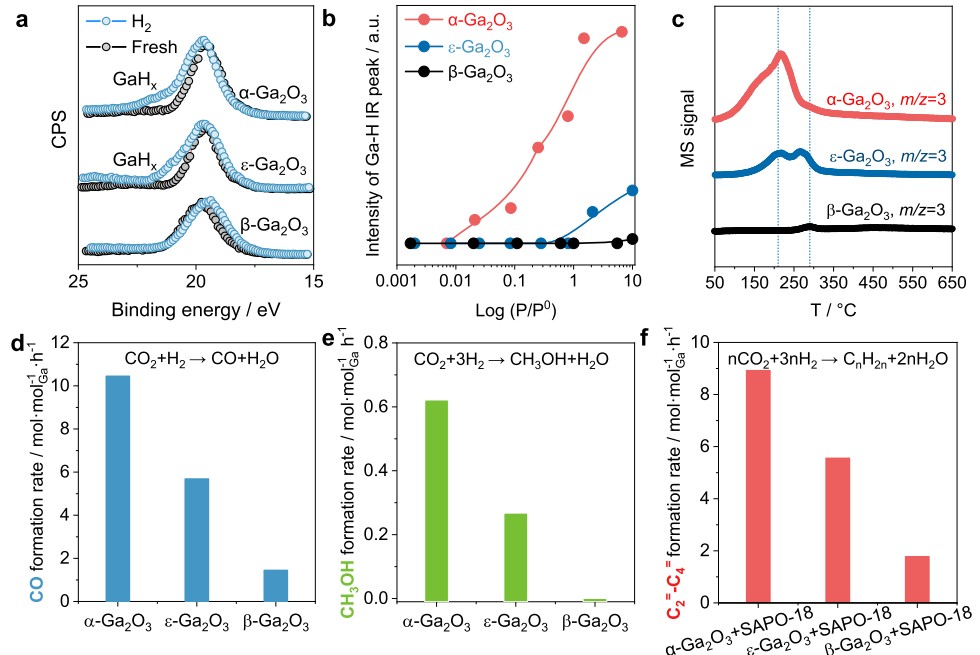

**Fig. 4 | Promotion effect of H₂ homolytic dissociation for CO₂ hydrogenation.** **a** In situ Ga *3d* XPS spectra of fresh $Ga_2O_3$ and $Ga_2O_3$ interacted with H₂ at 350 °C, 3 MPa. **b** H₂-chemisorption intensity of different $Ga_2O_3$ samples under different H₂ partial pressure measured by IR. **c** H₂-TPSR profiles with *m/z* = 3 (HD) signals in the effluents monitored by an online mass spectrometer. **d** CO formation rate over $Ga_2O_3$ during rWGS. Reaction conditions: H₂/CO₂ = 3 (*v/v*), 350 °C, 0.2 MPa,

9000 mL/g/h. **e** CH₃OH formation rate over $Ga_2O_3$ during methanol synthesis. Reaction conditions: H₂/CO₂ = 3 (*v/v*), 350 °C, 3 MPa, 6000 mL/g/h. **f** $C_2^=$-$C_4^=$ formation rate over $Ga_2O_3$ in OX-ZEO catalysis. Reaction conditions: OX/ZEO = 0.5 (mass ratio, 40–60 mesh), H₂/CO₂ = 3 (*v/v*), 350 °C, 3 MPa, 20,000 mL/g/h. Source data are provided as a Source Data file.

## Roles of activated hydrogen in CO₂ hydrogenation

The mechanism of CO₂ hydrogenation and the role of activated hydrogen species were further elucidated by a close look at the evolution of XANES and IR spectra during the transition between H₂ and CO₂. The catalysts were initially saturated with hydroxyl and hydride species by treatment with H₂ flow. Upon replacing H₂ with CO₂, various intermediates emerged immediately and exhibited growth over time, including bicarbonate species (HCO₃*) at 1630 cm⁻¹, formate species (HCOO*) at ~1386 and ~1578 cm⁻¹, and methoxy species (CH₃O*) at 2930, 1420, and 1475 cm⁻¹ (Fig. 5a, b and Supplementary Fig. 36)[10,40-42]. By tracking the time-resolved evolutions, it was observed that adsorbed CO₂ rapidly converted into HCO₃* by consuming surface hydroxyl groups on α-$Ga_2O_3$, while negligible HCO₃* was observed on β-$Ga_2O_3$. Fluctuations observed in Ga K-edge XANES indicated oxygen vacancies was formed on α-$Ga_2O_3$ upon exposure to H₂, which were subsequently eliminated upon introducing CO₂ (Fig. 5c and Supplementary Figs. 37–39). In contrast, the formation and consumption of oxygen vacancies were found to be difficult on β-$Ga_2O_3$ during the transitions between H₂ and CO₂, as the white line intensity of Ga K-edge in XANES remained unchanged. The results suggest that both hydroxyls and oxygen vacancies on α-$Ga_2O_3$ participate in adsorption and activation of CO₂, thereby providing α-$Ga_2O_3$ with a higher CO₂ adsorption capability compared to β-$Ga_2O_3$. This is supported by CO₂-TPD experiments (Supplementary Fig. 40).

Moreover, the in-situ IR experiments revealed the crucial promotion role of hydrides, as the hydrogenated CH₃O* species developed in a faster rate on α-$Ga_2O_3$ than on β-$Ga_2O_3$ (Fig. 5a, b). To understand this, the evolution of hydrides (Ga-H), hydroxyls (Ga-OD) and surface intermediates were monitored during cyclical switching of CO₂ and H₂/D₂ every half an hour at 350 °C. As shown in Fig. 5d, e and Supplementary Fig. 41, the bands of Ga-H and Ga-OD for both α-$Ga_2O_3$ and β-$Ga_2O_3$ appeared immediately upon the introduction of H₂ or D₂ into the cell, which then stabilized over time. When the carrier gas was replaced with CO₂, most Ga-H and half of Ga-OD

bands for α-$Ga_2O_3$ declined. The behaviors of Ga-H and Ga-OD species for β-$Ga_2O_3$ differ from that on α-$Ga_2O_3$. When β-$Ga_2O_3$ saturated with hydrides and hydroxyls was exposed to CO₂, Ga-H could be consumed, but Ga-OD remained intact (Fig. 5d, e). Importantly, when H₂ or D₂ was switched back, the bands of hydrides or deuterides regenerated, and the corresponding bands for HCOO* and CH₃O* gradually declined for both samples (Fig. 5f and Supplementary Fig. 42).

The Gibbs free energy profile of CO₂ hydrogenation on O-defective α-$Ga_2O_3$ (001) confirms this stepwise hydrogenation of CO₂* → HCOO* → H₂COOH* → CH₂O* → CH₃O* through reactions with surface hydrides (Supplementary Table 10). Moreover, HCOO* is found to accumulate on the surface as the reaction proceeds (Supplementary Fig. 43), indicating that the hydrogenation of HCOO* is the rate-determining step. This result is also supported by the slower formation rates of deuterated methoxy (CD₃O*) ($r_H/r_D$ of ~1.8) (Supplementary Fig. 44). In addition, model IR experiments for formate hydrogenation demonstrated that α-$Ga_2O_3$ can hydrogenate formate species to CH₃O*, while β-$Ga_2O_3$ remains inactive (Supplementary Fig. 45). Further semi-quantitative analysis of the IR peaks revealed a high CH₃O*/HCOO* ratio of ~1.5 for α-$Ga_2O_3$ and a relatively low ratio of 0.3-0.5 for β-$Ga_2O_3$ (Fig. 5f).

The role of highly active hydrides on α-$Ga_2O_3$ was further verified by in-situ XPS of catalysts that were treated in H₂, CO₂, and the CO₂-H₂ mixture at 3 MPa and 350 °C for 12 h (Supplementary Fig. 46). The shoulder of hydride-adsorbed Ga species disappeared, when the catalyst was exposed to CO₂ or the CO₂ + H₂ mixture, indicating the consumption of hydrides in the reaction with CO₂. Thus, with a detailed analysis of in-situ IR, XANES, and XPS, it was elucidated that α-$Ga_2O_3$ provides an abundant supply of highly active hydrides for kinetically relevant step of HCOO* to CH₃O* in CO₂ hydrogenation, in contrast to β-$Ga_2O_3$, which supplies limited hydrides. Above all, the intrinsic activity of the hydrides to hydrogenate CO₂ on octahedra Ga³⁺ of α-$Ga_2O_3$ is higher than that tetrahedral Ga³⁺ of β-$Ga_2O_3$ (Supplementary Fig. 47).

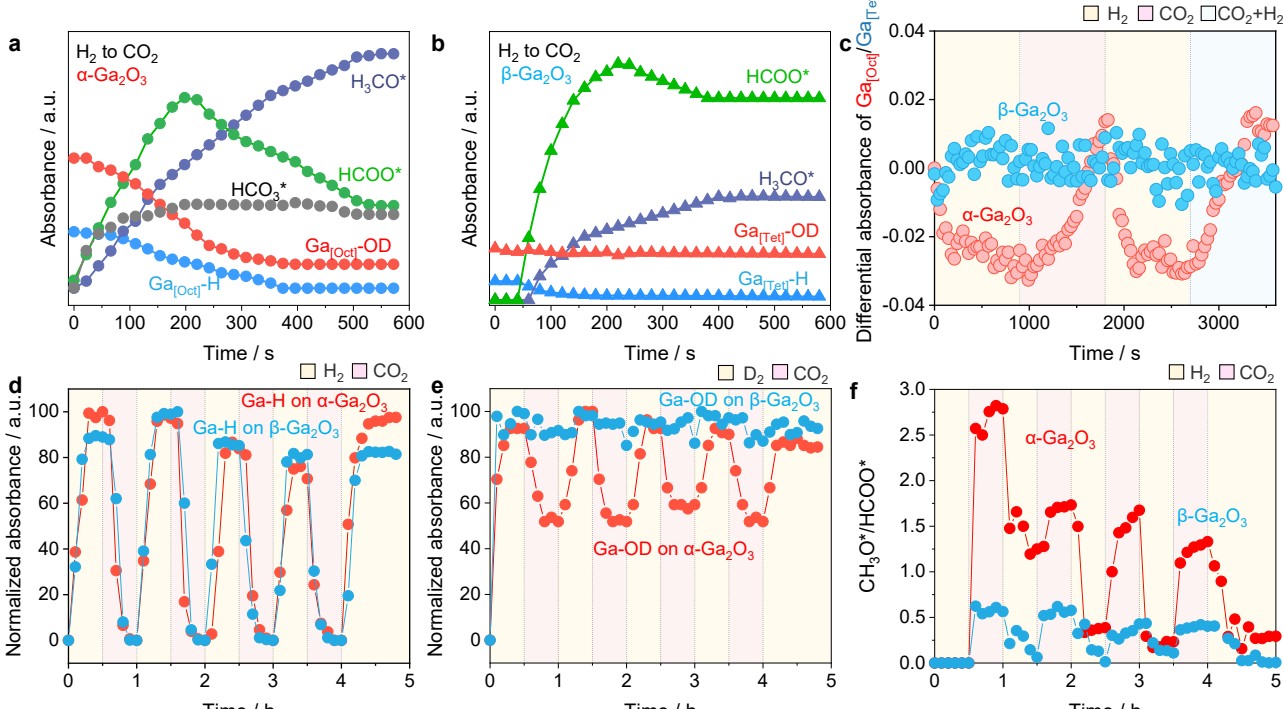

**Fig. 5 | Roles of activated hydrogen in CO₂ hydrogenation.** The time development of IR peak intensity of Ga-H, Ga-OD, HCO₃*, HCOO* and CH₃O* over α-Ga₂O₃ (**a**) and β-Ga₂O₃ (**b**). The evolution of hydroxyls was studied by using D₂, since the bands of OH are interfered by the overtone of CO₂ (3800−3500 cm⁻¹). **c** In-situ time-resolved X-ray absorption difference spectrum of Ga³⁺ in the near edge region of the Ga K-edge for Ga₂O₃ samples. The difference spectrum is obtained by subtracting Ar-treatment spectra. **d** The normalized IR peak intensity of Ga[Oct]-H over α-Ga₂O₃ and Ga[Tet]-H over β-Ga₂O₃ in H₂/D₂-CO₂ switching IR. **e** The normalized IR peak intensity of Ga[Oct]-OD over α-Ga₂O₃ and Ga[Tet]-OD over β-Ga₂O₃ in H₂/D₂-CO₂ switching IR. **f** The CH₃O*/HCOO* ratio over α-Ga₂O₃ and β-Ga₂O₃ in H₂/D₂-CO₂ switching IR. Source data are provided as a Source Data file.

Finally, we investigated whether the promotion of homolytic H₂ dissociation applies to other oxides containing octahedral Ga³⁺. Several binary spinel oxides including ZnGa₂O₄, MgGa₂O₄, and MnGa₂O₄ were studied. We used the calcination temperature to modulate the coordination structure of Ga³⁺ species, as the higher calcination temperature would regulate the inversion of spinel and increase the content of octahedral Ga³⁺ sites[38,39]. H₂-IR results show the different distributions of coordination of Ga³⁺ on the surfaces (Supplementary Fig. 48). Interestingly, we found that oxides with higher fractions of octahedral Ga³⁺ species exhibited superior yields and selectivity of light olefins (Supplementary Fig. 49). This indicates the importance of the presence of octahedral Ga³⁺ species in achieving enhanced performance in hydrogenation. Additionally, we also quantified surface hydrogen species over Cu/Ga₂O₃ (Supplementary Figs. 50−52). Cu/α-Ga₂O₃ was measured to have a much higher density of Ga-H than α-Ga₂O₃ and Cu/β-Ga₂O₃, which showed high activity to convert CO₂ into formate and methoxy intermediates (Supplementary Figs. 53-54). As a result, the methanol yield of Cu/α-Ga₂O₃ reached maximum of 0.4 mmol/g_cat/h at 300 °C, which is twice higher than that of Cu/β-Ga₂O₃ (Supplementary Figs. 55-56). This result proved that the α-Ga₂O₃ can serve as the promising material synergizing with metal to activate H₂ and further promote the CO₂ hydrogenation reaction.

## Discussion

In summary, we distinguished the homolytic and heterolytic H₂ dissociation pathway over Ga₂O₃ by the quantitative and time-resolved analyses of the complex dynamics during hydrogen activation. The formation of hydride species on α-Ga₂O₃, with a non-stoichiometric ratio to hydroxyls of 5.6 and a significantly higher formation rate compared to the heterolytic H₂ dissociation process, effectively demonstrates the occurrence of a homolytic dissociation pathway.

This homolytic H₂ cleavage benefits from a closer distance between Ga atoms over unsaturated Ga[Oct] metalloids, resulting in highly active and high-coverage hydrides. Cooperated with the oxygen vacancies and hydroxyls, the hydrides formed on α-Ga₂O₃ promote the adsorption of CO₂ and hydrogenation of key HCOO* intermediate, leading to enhanced CO₂ conversion and higher yield of valuable products including CO, methanol and light olefins. However, β-Ga₂O₃, which mostly consists of tetrahedral Ga³⁺ sites on the surface, fails to provide sufficient hydride and shows low activity for CO₂ hydrogenation. The homolytic hydrogen dissociation pathways and enhanced reactivity in CO₂ hydrogenation were also observed on several binary mixed oxides containing high fractions of octahedral Ga³⁺ species on the surfaces.

## Methods
### Catalyst preparation
Synthesis of Ga₂O₃: For α-Ga₂O₃, 6.0 g Gallium nitrate hydrate (Ga(NO₃)₂·xH₂O) was dissolved in 100 mL deionized water under the magnetic stirring. Liquid ammonia solution (NH₃·H₂O) was added into the clear solution until the pH was raised up to 9 and then stirred for 3 more hours. The precipitate was collected with filtering and dried overnight at 80 °C. Then, the sample was heated at 500 °C for 6 h in the air. The 6.0 g Ga(NO₃)₃·xH₂O was annealed in air at 500 °C for 6 h and white ε-Ga₂O₃ powder was obtained. For β-Ga₂O₃, 6.0 g Ga(NO₃)₃·xH₂O was dissolved into 150 mL anhydrous ethanol. Then, the Hydrazine monohydrate (H₄N₂·H₂O) was added into the solution dropwise under the magnetic stirring until the pH of the solution was raised up to 11. The precipitate was washed with ethanol, centrifugated and dried overnight at 80 °C. The resultant powder was finally calcined at 800 °C for 4 h in static air to obtain the sample.

Synthesis of SAPO-18 and MGa₂O₄. SAPO-18 samples were synthesized by treating an aluminophosphate-based gel hydrothermally[43,44].

7.4 g Aluminum hydroxide hydrate (55 wt% $Al_2O_3$) was added to a solution of 85% phosphoric acid in water (6.0 g). To this mixture, 4.8 g silica sol (30% $SiO_2$) and 8.3 g N, N-diisopropylethylamine ($C_8H_{19}N$) were added and a gel with the composition 1.6 $C_8H_{19}N$: 0.6 $SiO_2$: $Al_2O_3$: 0.9$P_2O_5$: 50$H_2O$ was formed by stirring vigorously[43,44]. Subsequently, this gel was sealed in a Teflon-lined stainless autoclave and heated at 180 °C for 7 days. The solid product was recovered by filtration, washed with distilled water and dried in air at 50 °C[43]. Finally, the as-synthesized sample was calcined in a stream of dry oxygen at 600 °C for 10 h[45]. Binary metal oxides including $ZnGa_2O_4$, $MgGa_2O_4$, and $MnGa_2O_4$ were synthesized by a coprecipitation method with the pH kept at 9 - 10. For $ZnGa_2O_4$, 2.97 g $Zn(NO_3)_2$·6$H_2O$ and 5.11 g $Ga(NO_3)_3$ • x$H_2O$ served as the precursors and their molar ratio was 1:2. For $MgGa_2O_4$, 2.56 g $Mg(NO_3)_2$·6$H_2O$ and 5.11 g $Ga(NO_3)_3$ • x$H_2O$ served as the precursors and their molar ratio was 1:2. For $MnGa_2O_4$, 3.58 mL 50 wt% aqueous solutions of $Mn(NO_3)_2$ and 2.56 g $Ga(NO_3)_3$ • x$H_2O$ served as the precursors and their molar ratio was 1:2. An aqueous solution of NaOH and NaHCO$_3$ with a molar ratio of 2:1 was used as the precipitant. After precipitation at 70 °C, the suspensions were aged for 2 h under continuous stirring. The precipitates were washed with deionized water until the pH of the supernatant was ca. 7, filtered and dried at 60 °C for 12 h, and then calcined at 300 and 600 °C for 4 h in air, respectively. The samples were denoted as $ZnGa_2O_4$-300, $MgGa_2O_4$-300, $MnGa_2O_4$-300, $ZnGa_2O_4$-600, $MgGa_2O_4$-600, and $MnGa_2O_4$-600 respectively[10]. The commercial sources and purities of utilized reagents were listed in Supplementary Table 11.

## Catalyst characterization

X-ray diffraction (XRD) of different $Ga_2O_3$ were recorded before and after reaction. XRD patterns were performed with 2$\theta$ values between 10° and 80° using a Bruker-D2 diffractometer employing the graphite filtered Cu K$\alpha$ radiation ($\lambda$ = 1.54056 Å). Particle size of $Ga_2O_3$ nanoparticles was calculated by Sherrer equation

$$D = \frac{k\lambda}{B \cos \theta} \qquad (1)$$

Where $k$ stands for the Sherrer constant, $\lambda$ is the wavelength of X-ray, $B$ is the full width at half maximum (fwhm) of the diffraction peak, and $\theta$ is the angle of diffraction. The specific surface areas of $Ga_2O_3$ were obtained on a Micromeritics Tristar 3000 analyzer by $N_2$ adsorption-desorption at 77 K. Prior to the tests, all catalysts were degassed at 300 °C for 3 h. The Brunauer-Emmett-Teller (BET) method was used to calculate the specific surface areas from the isotherms[46]. High-resolution transmission electron microscopy (HRTEM) images were obtained by a JEOL JEM−2100 electron microscope operated at an accelerating voltage of 200 kV. The samples were ultrasonically dispersed in ethanol and a drop of the solution was placed onto a copper grid coated with a thin microgrids support film. Annular dark field scanning TEM (ADF-STEM) was operated with an Oxford X-Max 80 SDD EDX detector at 200 kV.

The electronic structure of $Ga_2O_3$ was investigated by in-situ X-ray photoelectron spectroscopy (XPS). The binding energies were calibrated using the C 1s peak at 284.6 eV as a reference. Specifically, the catalysts were pre-treated in fixed-bed reactor, and then directly moved into the glove box atmosphere to load sample stage of XPS, which allows the sample transfer without exposing to air. The $CO_2$ hydrogenation reaction were performed under the following conditions: $CO_2$ (flow rate is 10 mL/min), $H_2$ (flow rate is 30 mL/min) and $CO_2/H_2$ mixture (ratio =1:3, flow rate is 40 mL/min) at 350 °C, 3.0 MPa. The atom ratio of O to Ga ($n_O$:$n_{Ga}$) was calculated as follow:

$$n_O : n_{Ga} = \frac{I_O/SF_O}{I_{Ga}/SF_{Ga}} \qquad (2)$$

Where $I_{Ga}$ represents the integration of Ga 3d peak area in XPS. $I_O$ represents the integration of the lattice O peak area in XPS. $SF_{Ga}$ represents the atomic sensitivity factor of Ga and $SF_O$ represents the atomic sensitivity factor of O.

Ga K-edge X-ray absorption near edge structure (XANES) experiments were undertaken at the BL05U beamline of the Shanghai Synchrotron Radiation Facility (SSRF), collecting in transmission mode using a Si (111) Bragg polychromator and a XIMEQ detector. 12 mg of the $Ga_2O_3$ was grinded for 30 min and pressed into a sheet with a diameter of 5.5 mm. The sheet was placed into an in-situ cell for test. The sample was then heated from room temperature to 350 °C under Ar flow. $CO_2$, $H_2$ or $H_2$ + $CO_2$ ($H_2/CO_2$ = 3 (v/v)) gas was purged into the reaction cell to carry out the in-situ experiments at 350 °C and 1.0 MPa. The energy dispersive mode of BL05U beamline enables the exposure time and acquisition time of X-ray absorption spectra to be 35 and 150 ms respectively. The XANES spectra were calibrated by $Ga_2O_3$ standard spectra.

The Fourier Transform Infrared Spectroscopy (FTIR, Thermo Scientific Nicolet IS50 spectrometer) was used to collect the vibrational spectra of adsorbed surface species under $CO_2$ hydrogenation atmosphere[45]. In-situ FTIR experiments were carried out in an in-situ transmission cell. About 15 mg $Ga_2O_3$ was pressed into sheets with a diameter of 13 mm and packed in the in-situ cell. All the samples were degassed at 500 °C under Ar flow (30 mL/min) for 3 h to obtain the clean surface, and the spectra of samples for each measurement were then collected by subtracting the background of blank cell[45]. Different gases can be switched quickly through 6-way valve. The absorbance normalized to the mass of the catalysts. Generally, the in-situ reactions were carried out under 1.0 MPa, 350 °C, 40 mL/min $CO_2$ + $H_2$ ($D_2$), and $CO_2$:$H_2$ ($D_2$) = 1:3[45].

Transient kinetic analysis with infrared spectroscopy (TKA-IR, $H_2(D_2)$-IR) were carried out at 350 °C with blank transmission cell as the background. All the samples were pre-treated at 550 °C in Ar flow (100 mL/min) for 5 h to obtain the flat and pure spectra. $H_2$ ($D_2$) was then switched quickly through 6-way valve, and the spectra of samples for contacting $H_2$ ($D_2$) were collected by subtracting the background spectrum. The spectra were recorded by collecting 32 scans at a resolution of 4 cm$^{-1}$. The intensity of infrared characteristic peak, such as Ga-H and Ga-OD, represents the number of the hydrogen-containing species. The outlet of the DRIFT cell was connected to online quadrupole mass spectrometer (MS), so the gas component in the effluent can be monitored and recorded by MS.

Temperature programmed surface reaction of $H_2$ ($H_2$-TPSR) was also performed on a Micromeritics AutoChem II 2920 apparatus connected to online MS. Typically, the $Ga_2O_3$ (100 mg) was dehydrated under Ar (30 mL/min) at 500 °C for 3 h. Then the dried samples were pretreated at 350 °C for 1 h under $D_2$ flow (15 mL/min) to generate the Ga-D and Ga-OD. After cooling to room temperature in Ar flow, a flow rate of 15 mL/min of $H_2$ was introduced until the MS signal was stable. The temperature was increased linearly from 50 °C to 800 °C with a rate of 10 °C·min$^{-1}$. The gas component in the effluent was monitored and recorded by mass spectrometer (PIMS 1500 Photo ionization Process TOF-MS, Jinkai Instrument (DaLian) Co., Ltd.). Typically, the mass/charge ratio ($m/z$) value is 3 for HD, which was produced via the reaction between Ga-D and $H_2$. The $m/z$ values are 2 for $H_2$ and 4 for $D_2$ in the hydrogen isotope analysis.

$H_2$-exchange experiment was also performed on a tubular reactor with quantitative ring (200 μL) connected to online mass spectrometer. Typically, the $Ga_2O_3$ was dehydrated under Ar (30 mL/min) at 500 °C for 3 h. Then the dried samples were pretreated at 350 °C for 1 h under $D_2$ flow (15 mL/min) to generate the Ga-D and -OD. After the temperature cooled to 150 °C in Ar flow, then 200 μL of $H_2$ in dosing ring was injected into the reactor cyclically until the MS signal of $H_2$ ($m/z$ = 2) is stable. Then the temperature increased to 350 °C in Ar flow, 200 μL of $H_2$ in dosing ring was injected again into the reactor cyclically

until the MS signal of $H_2$ (m/z = 2) is stable. The gas component in the effluent was monitored and recorded by MS. Combined with $H_2$-IR, the Ga-D and Ga-OD can be quantified.

$H_2$ chemisorption experiments were conducted with a Thermo Scientific Nicolet IS50 spectrometer connected with vacuum system. In a typical procedure, 15 mg of the sample was degassed at 200 °C in Ar and evacuated at 350 °C for 30 min. Prior to the chemisorption experiments, the sample was further evacuated for 40 min. The adsorbates ($H_2$) were introduced into the system for the measurements of chemisorption isotherms. The first chemisorption isotherm was measured in the pressure range of 0.002-10 mbar at 350 °C.

Temperature programmed reduction of $H_2$ ($H_2$-TPR) was performed on a Micromeritics AutoChem II 2920 apparatus. Typically, the catalyst (50 mg) was pretreated at 200 °C for 1 h under Ar flow for dehydration. After cooling to room temperature, a flow rate of 15 mL/min of $H_2$ was introduced for the $H_2$ reduction. Then the temperature was increased linearly from 50 °C to 600 °C with a rate of 10 °C·min⁻¹. The gas component in the effluent was monitored and recorded by online quadrupole mass spectrometer. Temperature programmed desorption of $H_2$ ($H_2$-TPD) was performed on a Micromeritics AutoChem II 2920 apparatus. Typically, the catalyst (50 mg) was pretreated at 350 °C for 1 h under 10% $H_2$/Ar flow (30 mL/min) to reduce the catalysts. After cooling to room temperature, the purging was carried out by 30 mL/min of Ar for 1 h. Then the temperature was increased linearly from 50 °C to 600 °C with a rate of 10 °C·min⁻¹ [47]. The gas component in the outlet was monitored by online mass spectrometer[47]. Temperature programmed desorption of $CO_2$ ($CO_2$-TPD) was also performed on a Micromeritics AutoChem II 2920 apparatus. Typically, the catalyst (50 mg) was pretreated at 350 °C for 1 h under 10% $H_2$/Ar flow (30 mL/min) to activate the catalysts. After cooling to room temperature, $CO_2$ adsorption was carried out under a flow rate of 30 mL/min of $CO_2$[47]. Subsequently, the purging was carried out at the same temperature by 30 mL/min of Ar for 1 h[47]. Then the temperature was increased linearly from 50 °C to 600 °C with a rate of 10 °C·min⁻¹. The gas component in the outlet was monitored by a thermal conductivity detector (TCD).

### Catalytic testing

All experiments were performed in a glass-lined stainless steel reactor tube (8 mm inner diameter) under a continuous flow. Typically, 300 mg composite catalyst (40-60 meshes) with oxide/zeolite=1/2, 2/1 and 4/1(mass ratio) was placed in the reactor. Ar was added after the back pressure valves as the internal standard for online gas chromatography (GC) analysis. Reaction was carried out under conditions: $H_2$/$CO_2$ = 3 (v/v), 350 °C, 3.0 MPa, gas hourly space velocity (GHSV) = 2000−20000 mL/g/h. At the outlet of the reactor, the gases were decompressed to the atmospheric pressure and the reaction products were analyzed with online gas chromatograph (Agilent 8890), equipped with a TCD and two flame ionization detectors (FID). Oxygenates and hydrocarbons up to $C_{12}$ were analyzed by FID with HP-FFAP and HP-AL/S column. A thermal conductivity detector with columns of Hayesep Q and 5 A molecular sieves packed columns for other gaseous product (including $H_2$, CO, $CO_2$ and $CH_4$) was used. $CH_4$ was taken as a reference bridge between FID and TCD. $CO_2$ conversion, C-containing products selectivity, and yield of products were calculated as follow:

$$X_{CO_2}(\%) = \frac{F_{CO_2,in} - F_{CO_2,out}}{F_{CO_2,in}} \times 100 \qquad (3)$$

$$S_N(\%) = \frac{\%N}{\sum((\%N))} \times 100 \qquad (4)$$

$$Yield_N(\%) = X_{CO_2} \times S_N(\%) \times 100 \qquad (5)$$

where $F_{CO_2,in}$ and $F_{CO_2,out}$ in Eq. (3) represented moles of $CO_2$ at the inlet and outlet, respectively. Where N in Eq. (4) represents the carbon-containing species in all of the products. The selectivity of individual hydrocarbon $C_nH_m$ ($Sel_{CnHm}$) among hydrocarbons (free of CO) in Eq. (6) was calculated according to

$$Sel_{CnHm}(\%) = \frac{nCnHm}{\sum(nCnHm)} \times 100 \qquad (6)$$

The results were obtained when the reaction had reached a steady state. The carbon balance was found to be > 95% in all the tests.

### DFT calculation

All DFT calculations are performed via the plane wave VASP code[48], where electron-ion interaction is represented by the projector augmented wave pesudopotential[49–51]. The exchange functional utilized is the spin-polarized GGA-PBE[52]. The kinetic energy cutoff is set as 450 eV[53]. The first Brillion zone k-point sampling utilizes the Monkhorst-Pack scheme with an automated mesh determined by 18 times of the reciprocal lattice vectors[53]. The energy and force criterion for convergence of the electron density and structure optimization are set at $10^{-6}$ eV and 0.05 eV/Å, respectively[53].

Ab-initio thermodynamics analyses. To determine the equilibrium $O_v$ concentration in $Ga_2O_3$ surfaces, the ab-initio thermodynamics analyses have been performed where the formula

$$Ga_xO_z + nH_2 \rightarrow Ga_xO_{z-n} + nH_2O \qquad (7)$$

are used to compute the free energy change as a function of temperature and $H_2$ partial pressure. To determine the Gibbs free energy change ($\Delta G$) per formula unit (f. u.) for the above reactions, one needs to compute

$$\Delta G(p,T) = G[GaO_x](p,T) + (y-x)*\mu[H_2O](p,T)$$
$$- G[GaO_y](p,T) - (y-x)*\mu[H_2](p,T) \qquad (8)$$

where G is the Gibbs free energy of surfaces and $\mu$ is the chemical potential for molecules. The G[X] can be approximated by their DFT total energy E[X] with appropriate inclusion of zero-point-energy (ZPE), since it is known that the vibration entropy and the pV term contributions of solid phases are negligibly small. The chemical potential for molecules $\mu[X]$ can be calculated as follows:

$$\mu[X](p,T) = E[X] + ZPE[X]$$
$$+ \left[ H[X](p^0,T) - H[X](p^0,0K) - TS[X](p^0,T) + k_BT\ln\frac{p}{p^0} \right] \qquad (9)$$

where enthalpy (H) and entropy (S) terms are taken from the standard thermodynamics data.

## Data availability
All data supporting the findings of this study are available within the paper and its supplementary information files. Source data are provided with this paper.

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

## Acknowledgements

This work was supported by the Ministry of Science and Technology of China (2022YFA1503804, Y.Z.), National Natural Science Foundation of China (22102033, 22272031, Y.Z.), Science & Technology Commission of Shanghai Municipality (22ZR1408000, 22QA1401300, Y.Z.), China National Postdoctoral Program for Innovative Talents (BX20220090, C.S.Y.), China Postdoctoral Science Foundation (2022M710740, C.S.Y.), the Fundamental Research Funds for the Central Universities (20720220008, Y.Z.) and the Foundation of Key Laboratory of Low-Carbon Conversion Science & Engineering (KLLCCSE-202201, Y.Z.). We are grateful to all staff at the BL05U and BL06B beamline of the Shanghai Synchrotron Radiation Facility, Shanghai Advanced Research Institute, Chinese Academy of Sciences.

## Author contributions

C.S.Y. performed most material synthesis, characterization, and catalytic tests. C.S.Y., Y.Z. and X.B. initiated the project and wrote the manuscript. S.M. and Z.-P.L. performed the theoretical calculations and helped the manuscript drafting. F.Y., W.-P.S., and L.Z. helped with the XPS experiments. L.W. helped with the XAS experiments. D.Y. helped with the catalytic tests and IR experiments. T.L., Y.C. and Y.L. provided constructive suggestions for the work and helped with the temperature-programmed experiments. H.H. and H.D. helped with H1 NMR measurement and analysis.

## Competing interests

The authors declare no competing interests.
