## [Peer Review File · Nature Communications]

Homolytic H₂ dissociation for enhanced hydrogenation catalysis on oxidesREVIEWER COMMENTS

Reviewer #1 (Remarks to the Author):

In this manuscript, Yang et al. reported a comprehensive study on hydrogen activation over three types of Ga₂O₃. They intended to quantify the contribution of heterolytic and homolytic dissociation of H₂ and further correlate it with the catalytic performance in CO₂ hydrogenation. Overall, this work is quite interesting and may generate high impact to the field if they can further strengthen their claims. More detailed comments are provided below.

1. The starting point of the postulation of homolytic H₂ dissociation is the quantification of proton and hydride, which was based on the IR assignments of GaO-H/GaO-D and Ga-H/Ga-D bands and the MS quantification of H₂ consumption (or HD formation? This does not seem to be clear). The authors should provide full spectra of their IR results in order to give readers a complete picture. Their IR spectra were acquired and reported in differential format with the samples themselves in Ar atmosphere at 350C as background. Differential IR spectra sometimes give misleading conclusions if the baseline is not flat. Their assigned Ga-D band in Figs 2b and 2d apparently shows a negative band on the higher wavenumber side, which becomes flat upon H₂ treatment at 350C. This feature (negative peak on the left, positive band on the right) is likely associated with the redshift of an existing IR band of Ga₂O₃ by switching atmosphere from Ar to D₂. Without seeing the full spectra, it is hard to justify their band assignments. On the other hand, is it possible to measure the IR spectra with KBr or other background to show the actual IR features?
2. Do they have a more explicit way to quantify the amount of newly formed proton and hydride upon exposure in H₂? How about NMR?
3. Fig S13 (b, d, e) shows an increasing H₂ signal but the HD signal first increases and then drops. It seems that H₂ was consumed without forming HD at the beginning. What caused the delay in HD formation?
4. The DFT results discussed on page 8 suggest that the initially heterolytically dissociated hydroxyl/hydride pair tends to evolve into two hydroxyls. They computed the homolytic H₂ dissociation in the presence of adjacent hydroxyls. The overall result is similar to two heterolytically dissociated H₂ (two hydroxyls and two hydrides). The DFT results do not seem to support the claim of excess hydride formation.

Reviewer #2 (Remarks to the Author):

The manuscript deals with the role of heterolytic splitting in hydrogenations on oxides. While most of the results rely on the computational simulations these are kept to the minimum and the real important data are neither presented nor discussed. I find that the manuscript is below the quality required for Nature Comm. and therefore I suggest submission to a more specific journal.

1. I would expect a full characterization of the three material phases. This shall include at least their

corresponding structures, low lying surfaces, Bader charges and Density of States for all three phases and the careful comparison to literature and experimental values.

2. Once hydroxylated the surface might change, again I would expect a full characterization.
3. The TS characterization needs to include the transition states vibrational characterization.
4. All the structures need to be uploaded to a suitable database for other researchers to reproduce the data found here.
5. As points 1-3 are crucial to understand why the different systems behave so differently I cannot recommend the work for publication.

Reviewer #3 (Remarks to the Author):

This is an very interesting work with detailed spectroscopy evidence for the presence of two H₂ dissociation channels on Ga₂O₃ and how they are impacted by the phase/surface structure of Ga₂O₃. The finding was also used to explain the large catalytic differences observed in several hydrogenation reactions over different Ga₂O₃ phases. I commend the authors for a very careful investigation on the fundamentals of H₂ interaction and reaction over an oxide surface. The work can be accepted for publication after some revisions as noted below.

1) is the intrinsic activity (i.e., TOF) of the hydrides on octahedra Ga³⁺ the same for the three different phases? or is it depend on the Ga₂O₃ phase? The IR switch experiment results could be used to derive the intrinsic rates of the surface hydrides on alpha and beta-Ga₂O₃.

2) it is interesting to compare the reactivity of the surface hydrides on Ga₂O₃ in the hydrogenation reactions with Ga₂O₃ supported metal catalysts (such as Cu for CO₂ to methanol or CO). This will provide an idea of how reactive these hydrides are in comparison to hydrogenation assisted with a metal surface.

3) a scheme should be provided on how the 2nd dissociation channel, the homolytic dissociation, occurs and the migration of H to form extra hydride.

4) some of the claims/sentences need be clarified. for examples, on P3 Line 34-35: "breaking the ceiling of hydrogenation capacity for oxides", what is the ceiling of hydrogenation capacity of oxides? I think the authors meant the capacity of surface hydrogen is above the ceiling of a monolayer H on an oxide surface? on α-Ga₂O₃, the authors claimed a 1.6 ratio for H/surface Ga³⁺, what is the ratio of H/surface O? is the total H (hydride + proton (OH)) density higher than the total surface atom density (Ga+O)? Anyway, the sentence needs revision and more relevant discussion is warranted in later section.

Reviewer #1:

Comment: In this manuscript, Yang et al. reported a comprehensive study on hydrogen activation over three types of Ga₂O₃. They intended to quantify the contribution of heterolytic and homolytic dissociation of H₂ and further correlate it with the catalytic performance in CO₂ hydrogenation. Overall, this work is quite interesting and may generate high impact to the field if they can further strengthen their claims. More detailed comments are provided below.

Response: We appreciate the reviewer for his/her carefully reading and the professional comments. In the revised version, we have made all the additions of experiments and discussions (highlighted with yellow background in the revised manuscript and supporting information), as suggested by the reviewer. The modification strengthens the conclusion of homolytic dissociation of H₂ on α-Ga₂O₃ in this work. Our point-by-point responses are listed below.

Comment 1-1): The starting point of the postulation of homolytic H₂ dissociation is the quantification of proton and hydride, which was based on the IR assignments of GaO-H/GaO-D and Ga-H/Ga-D bands and the MS quantification of H₂ consumption (or HD formation? This does not seem to be clear).

Response: We thank the reviewer for pointing out the unclear expressions. We have revised the corresponding parts for better clarification.

In our study, the quantification of proton and hydride was based on the MS signal of **H₂ consumption** during H₂-D₂ exchange experiments. The mass spectrometry (MS) signal was calibrated with 200 μL/pulse H₂. The attenuation of MS signals (m/z=2) during H₂-D₂ exchange experiments represents the consumption of hydrogen participating in the exchange reaction. The experiments ensure a reliable quantification of surface Ga-H and Ga-OH. We have bolded and marked the right vertical axis of **Figure 2c** and **Figure 2e** in red to make the expression clear. Additionally, more detailed description on the quantitative process has been added in revised **Supporting information**.

‘The difference between A (Stable) and A_i represents the consumption of hydrogen participating in the exchange reaction.’

Revised Figure 2. Quantitative analysis of hydride and hydroxyl species on Ga₂O₃ surface. c TKA-MS (Blue) and the amount of H₂ consumption (Red) during H₂-exchange experiment at 150 °C after α-Ga₂O₃ were saturated with D₂ at 350 °C. e TKA-MS (Blue) and the amount of H₂ consumption (Red) during H₂-exchange at 350 °C after α-Ga₂O₃ were saturated with D₂ at 350 °C.

Comment 1-2): The authors should provide full spectra of their IR results in order to give readers a complete picture.

Response: We appreciate the reviewer’s valuable suggestion. In response to this comment, we have provided full spectra for all of the IR results, including H₂-reduction process, TKA-IR and H₂-D₂ exchange experiment. The full IR spectra were supplemented with

blank transmission cell as the background. These IR spectra were updated in Supporting information (Figure R1-R3, Figure S6, S15 in Supporting information).

Figure R1. **a** *In-situ* FTIR of α -Ga₂O₃ samples contacting Ar to H₂ at 350 °C. **b** *In-situ* FTIR of β -Ga₂O₃ samples contacting Ar to H₂ at 350 °C.

Figure R2 (Figure S6 in Supporting information). **a** *In-situ* FTIR of α -Ga₂O₃ samples contacting H₂ to D₂ at 350 °C, 1 MPa. **b** *In-situ* FTIR of β -Ga₂O₃ samples contact H₂ to D₂ at 350 °C, 1 MPa.

Figure R3 (Figure S15 in Supporting information). The full IR spectrum with blank transmission cell as the background. **a** IR spectra of α - Ga_2O_3 during H_2 -exchange at 150 °C after the catalysts were saturated with D_2 at 350 °C. **b** IR spectra of α - Ga_2O_3 during H_2 -exchange at 350 °C after the catalysts were saturated with D_2 at 350 °C. **c** IR spectra of ϵ - Ga_2O_3 during H_2 -exchange at 150 °C after the catalysts were saturated with D_2 at 350 °C. **d** IR spectra of ϵ - Ga_2O_3 during H_2 -exchange at 350 °C after the catalysts were saturated with D_2 at 350 °C. **e** IR spectra of β - Ga_2O_3 during H_2 -exchange at 150 °C after the catalysts were saturated with D_2 at 350 °C. **f** IR spectra of β - Ga_2O_3 during H_2 -exchange at 350 °C after the catalysts were saturated with D_2 at 350 °C.

Comment 1-3): Their IR spectra were acquired and reported in differential format with the samples themselves in Ar atmosphere at 350C as background. Differential IR spectra sometimes give misleading conclusions if the baseline is not flat. Their assigned Ga-D band in Figs 2b and 2d apparently shows a negative band on the higher wavenumber side, which becomes flat upon H_2 treatment at 350C. This feature (negative peak on the left, positive band on the right) is likely associated with the redshift of an existing IR band of Ga_2O_3 by switching atmosphere from Ar to D_2 .

Response: We thank the reviewer for the helpful advice. Indeed, the differential format of

IR spectra might give misleading conclusions. As suggested by the reviewer, we have supplemented the full IR spectrum with blank transmission cell as the background (**Figure 2b and 2d, Figure R1-R3**). The IR spectra with the blank transmission cell as background have preserved the structural information of the sample itself. The negative band observed previously from the differential IR spectra might be attributed to the slight background shift, as shown in the **Figure R1**.

Revised Figure 2. Quantitative analysis of hydride and hydroxyl species on Ga_2O_3 surface. **b** IR spectra of $\alpha\text{-Ga}_2\text{O}_3$ during H_2 -exchange at 150°C after the catalysts were saturated with D_2 at 350°C . **d** IR spectra of $\alpha\text{-Ga}_2\text{O}_3$ during H_2 -exchange at 350°C after the catalysts were saturated with D_2 at 350°C .

The IR signal of blank transmission cell was used as the background. The experimental operations are as follows. Firstly, the samples were treated in *in situ* transmission cell under high-flow-rate Ar flow (100 ml/min) at 550°C for more than 5h before TKA-IR. After this

treatment, the clean surface was obtained. Then, the samples were cooled to the target temperature of 350°C for our measurements. The IR spectra collected with a blank *in situ* transmission cell as background displayed as almost a straight line, verifying the clean surface of samples. As shown in **Figure R1**, there appeared two additional IR peak at ca. 1980 cm⁻¹ and 3700 cm⁻¹ upon H₂ treatment at 350°C, which are attributed to proton (Ga-H) and hydride (Ga-OH) respectively. Except for these two peaks, no other surface species peaks were observed. The quantification of the surface proton and hydride in the new IR spectra (TKA-IR and H₂-D₂ exchange experiments) also confirmed the finding of homolytic H₂ dissociation on α-Ga₂O₃.

Figure R1. **a** *In-situ* FTIR of α-Ga₂O₃ samples contacting Ar to H₂ at 350 °C. **b** *In-situ* FTIR of β-Ga₂O₃ samples contacting Ar to H₂ at 350 °C.

We have also included detailed explanations for the negative band in **Figure 2b and 2d** of previous manuscript. This negative peak may be attributed to background shift induced by an existing IR band of Ga₂O₃. Previously, we used differential format with the samples themselves in Ar atmosphere as background. This operation caused significant fluctuations in the spectrum even due to trace background perturbation on the surface. The fluctuations also covered up the information of surface species. As the reviewer suggested, using a blank cell as the background would solve this problem. When we followed the suggestions of the reviewer and conducted sufficient degassing on the sample, the structure of Ga₂O₃ was stabilized, and the IR spectrum became flat. As a result, this background

shift (negative peak) became weak in IR spectra with blank transmission cell as the background. Most notably, the band assignments and the evolution of surface proton/hydride in the full IR spectra are in alignment with that in differential IR spectra.

We thank the reviewers for providing us with valuable guidance and we have added the new IR spectra (revised **Figure 2**) in the revised manuscript.

Comment 1-4): Without seeing the full spectra, it is hard to justify their band assignments. On the other hand, is it possible to measure the IR spectra with KBr or other background to show the actual IR features?

Response: We are grateful to the reviewer for the professional advice. **The full IR spectrum with blank transmission cell as the background** has been supplemented to preserve the species originally contained in Ga_2O_3 .

Figure R2 (Figure S6 in Supporting information). **a** *In-situ* FTIR of $\alpha\text{-Ga}_2\text{O}_3$ samples contacting H_2 to D_2 at $350\text{ }^\circ\text{C}$, 1 MPa. **b** *In-situ* FTIR of $\beta\text{-Ga}_2\text{O}_3$ samples contact H_2 to D_2 at $350\text{ }^\circ\text{C}$, 1 MPa.

Similarly, in full IR spectra of $\text{H}_2\text{-D}_2$ exchange experiments with blank transmission cell as the background ((**Figure R2**, **Figure S6 in Supporting information**), (**Figure R3**, **Figure S15 in Supporting information**)), the gradual conversion of hydrides (Ga-H at ca. 1980 cm^{-1} , Ga-OH at ca. 3700 cm^{-1}) to deuterides were observed (Ga-D at ca. 1430 cm^{-1} , Ga-

OD at ca. 2700 cm^{-1}). The IR peak position ratio of hydrides to deuterides (Ga-H/Ga-D, Ga-OH/Ga-OD) is about 1.4, confirming the assignment of hydride and deuteride in IR spectra. Moreover, there are no other IR peaks were observed except for hydrides and deuterides. This result suggested that TKA-IR experiments were performed over a clean surface. We believe that the band assignments are confirmed in the full infrared spectrum we provided.

Page S5 in Supporting information:

‘Transient kinetic analysis with infrared spectroscopy (TKA-IR, $\text{H}_2(\text{D}_2)$ -IR) reactions were carried out at 350 °C with blank transmission cell as the background. All the samples were pre-treated at 550 °C in Ar flow (100 mL/min) for 5 h to obtain the flat and pure spectra.’

Figure R3 (Figure S15 in Supporting information). The full IR spectrum with blank transmission cell as the background. **a** IR spectra of $\alpha\text{-Ga}_2\text{O}_3$ during H_2 -exchange at 150 °C after the catalysts were saturated with D_2 at 350 °C. **b** IR spectra of $\alpha\text{-Ga}_2\text{O}_3$ during H_2 -exchange at 350 °C after the catalysts were saturated with D_2 at 350 °C. **c** IR spectra of $\epsilon\text{-Ga}_2\text{O}_3$ during H_2 -exchange at 150 °C after the catalysts were saturated with D_2 at 350 °C. **d**

IR spectra of ϵ -Ga₂O₃ during H₂-exchange at 350 °C after the catalysts were saturated with D₂ at 350 °C. **e** IR spectra of β -Ga₂O₃ during H₂-exchange at 150 °C after the catalysts were saturated with D₂ at 350 °C. **f** IR spectra of β -Ga₂O₃ during H₂-exchange at 350 °C after the catalysts were saturated with D₂ at 350 °C.

Comment 2): Do they have a more explicit way to quantify the amount of newly formed proton and hydride upon exposure in H₂? How about NMR?

Response: We appreciate the valuable suggestion regarding the quantification of newly formed proton and hydride. We have conducted *Quasi in-situ* NMR to quantify the amount of newly formed proton and hydride, which confirmed the homolytic dissociation of hydrogen on α -Ga₂O₃.

***Quasi in-situ* NMR:**

The samples after hydrogen reduction at 350°C for 1h were transferred into the N₂-filled glove-box and then sealed into the NMR tube, followed by the ¹H-spectrum test. This operation keeps the sample out of air and ensures the surface structure of samples unchanged. We also conducted the measurements on the reference Ga₂O₃ treated in Ar flow at 350°C for 1h. The NMR results also suggest that the homolytic dissociation of H₂ takes place on α -Ga₂O₃.

As shown in H¹ NMR of **Figure R4**, two NMR signals appeared for Ga₂O₃ after H₂ treatment at 350°C. According to previous literature (*J. Phys. Chem. B*, **2006**, *110*, 5498-5507., *Chem. Eur. J.*, **2014**, *20*, 4038-4046., and *J. Am. Chem. Soc.*, **2022**, *144*, 17365-17375.), the NMR signal of 14.2 ppm can be attributed to the Ga-H NMR signature. This assignment is supported by the absence of 14.2 ppm on the reference sample treated in Ar at 350°C. The NMR signal at 0.2 ppm is belonging to the terminal hydroxyl group (μ_1 -OH), while 1.3, 1.9, 3.2, and 5.4 ppm are assigned to doubly bridging (μ_2 -OH) and triply bridging hydroxyl group (μ_3 -OH) on Ga₂O₃, respectively. We found that NMR signal intensity of Ga-H for α -Ga₂O₃ is much higher than that of β -Ga₂O₃, while the signal intensity of -OH is similar

for both two samples (**Figure R4**).

We then quantified the corresponding NMR peaks of newly formed Ga-H and Ga-OH during H₂ reduction by an external standard method. Adamantane was used as the external standard both for the chemical shift and the quantification of the ¹H molar number corresponded to related peaks. The amount of Ga-OH formed during H₂ reduction is calculated by subtracting the amount of hydroxyl groups on Ar-treated sample from the hydroxyl groups on reduced samples (**Figure R5b**). The ratio of Ga-H/Ga-OH over α-Ga₂O₃ is 3.2 while that of β-Ga₂O₃ is 0.5, confirming the homolytic dissociation of H₂ on α-Ga₂O₃ (**Figure R5a**).

The samples after Ar-treated may experience surface dehydration, resulting in a lower amount of -OH. Thus, the amount of -OH formed during H₂ reduction, calculated by deducting the amount of -OH on Ar-treated sample, is higher than the actual value. Accordingly, the actual Ga-H/Ga-OH over α-Ga₂O₃ formed during H₂ reduction should be higher than 3.2.

It is worth noting that Ga-H and Ga-OH would be gradually desorbed during the process of degassing and drying through in situ infrared spectroscopy (**Figure R5c**). After three hours of degassing, one-third of the surface Ga-OH is desorbed, and half of the surface Ga-H is desorbed. Therefore, the accurate quantification can not be made by the *quasi in-situ* NMR, due to the time-consuming transfer processes including the cooling down, sample loading in glove boxes, transfer, and other steps. Most of surface hydrogen species would be gradually desorbed before performing NMR. These experiments also highlight the advantages of TKA-IR and TKA-MS, which achieved *operando* quantification of surface H-species.

Figure R4. **a** ^1H MAS NMR spectra of $\alpha\text{-Ga}_2\text{O}_3$ samples after H_2 activation at 350°C (red lines), and after degassing under Ar flow at 350°C (blue lines), recorded at 9.4 T and MAS rate of 20 kHz. **b** ^1H MAS NMR spectra of $\beta\text{-Ga}_2\text{O}_3$ samples after H_2 activation at 350°C (red lines), and after degassing under Ar flow at 350°C (blue lines), recorded at 9.4 T and MAS rate of 20 kHz.

Figure R5. **a** The amount of Ga-H and Ga-OH formed after H_2 activation at 350°C measured based on ^1H MAS NMR spectra. **b** The amount of Ga-OH formed during H_2 activation at 350°C measured based on ^1H MAS NMR spectra. **c** The IR spectra of $\alpha\text{-Ga}_2\text{O}_3$ samples during degassing process for 3 h under Ar flow.

Comment 3): Fig S13 (b, d, e) shows an increasing H_2 signal but the HD signal first increases and then drops. It seems that H_2 was consumed without forming HD at the beginning. What caused the delay in HD formation?

Response: We appreciate the reviewer's meticulous observation and rigorous logic. We are grateful to the reviewer for pointing out this peculiar phenomenon and helping us with

insight into the dissociation mechanism of H₂.

We analysed the H₂-D₂ exchange reaction and divided this reaction into three separate processes. They are sequentially H₂ adsorption, surface exchange reaction between H₂ and Ga-D and HD desorption processes.

We have conducted isothermal adsorption experiments, H₂-TKA and H₂-TPD for the three processes to understand this delay phenomenon.

(1) H₂ adsorption

Firstly, the characterization of H₂ adsorption was obtained through isothermal adsorption experiments at 350°C. The adsorption isotherm curve shows that H₂ adsorption for α-Ga₂O₃ quickly reached saturation under relative low pressure at 350°C. In addition, the adsorption coefficients of H₂ for α-Ga₂O₃ and β-Ga₂O₃ are 1.8 and 0.4, respectively (**Figure R6**). As a result, the quick and strong H₂ adsorption for α-Ga₂O₃ were confirmed.

Figure R6. Isotherm adsorption of H₂ and adsorption coefficient for α-Ga₂O₃ and β-Ga₂O₃ at 350°C.

(2) Surface exchange reaction

The more detailed mechanism analysis was conducted to explore other possible reasons for the delay. We carried out H₂-TKA experiments to confirm this phenomenon. After the analysis of results, the delay phenomenon is speculated to root in a mechanism similar to the Bimolecular Nucleophilic Substitution Reaction (S_N2, Ga-D+H₂→Ga-H+HD) over α-Ga₂O₃, which is proven as below.

Firstly, we conducted the D₂ to Ar to H₂ switch experiments to get the information of H₂ conversion and HD formation over time (**Figure R7**). We found the HD formed through H₂-exchange with Ga-D was slow over α-Ga₂O₃. The entire detection process of HD over α-Ga₂O₃ lasted for 300 seconds, while HD on β-Ga₂O₃ was completely formed within 50 seconds. However, H₂ reacted with Ga-D and was consumed only during the initial stage of 50 seconds for all samples. These results confirmed the delay phenomenon on α-Ga₂O₃. Also, we performed differential processing on the H₂ mass spectrometry signal (m/z=2) over time to obtain the conversion rate of H₂ (**Figure R8**). The formation of HD was found slower than the consumption of H₂ over α-Ga₂O₃. In other words, the two molecules (H₂ and Ga-D) are speculated to be combined first before the cleavage of old chemical bonds. At atomic scale, the formation of new chemical bonds (H₂-D-Ga) may occur initially and the cleavage of old chemical bonds may occur subsequently (breaking of H-H and Ga-D). Also, the combination of H₂ and Ga-D is speculated to be the rate-controlling step in surface exchange reaction of H₂-D₂. We believe that the exchange reaction may follow the mechanism of H-D bond formation followed by Ga-D bond breaking, causing the delay in HD formation.

The possible pathway of H₂-D₂ exchange reaction is described as below:

Figure R7. D₂-H₂ switch experiments at 350 °C from D₂ to Ar and then to H₂. (a) α -Ga₂O₃ (b) ϵ -Ga₂O₃, and (c) β -Ga₂O₃.

Figure R8. H₂ conversion rate and HD formation rate in D₂-H₂ switch experiments at 350 °C from D₂ to Ar and then to H₂. (a,d) α -Ga₂O₃ (b,e) ϵ -Ga₂O₃, and (c,f) β -Ga₂O₃.

(3) HD desorption

The characterization of H₂ (HD) desorption was obtained through H₂-TPD experiments. The higher temperature of H₂-desorption on α -Ga₂O₃ (above 350°C) was observed in **Figure R9**. As a result, the slow desorption for H₂ (same as HD) even at higher temperature (350°C) for α -Ga₂O₃ were confirmed.

Figure R9. H₂-TPD profile of α -Ga₂O₃, ϵ -Ga₂O₃, and β -Ga₂O₃.

Overall, the details about the dissociation mechanism of H₂ at atomic scale are required through further characterization. Here, we have provided a possible hypothesis based on several preliminary experiments. We reckon the potential mechanism of H-D bond formation followed by Ga-D bond breaking and strong adsorption of H₂(HD) on α -Ga₂O₃ caused the delay in MS signal of HD.

***Comment 4):** The DFT results discussed on page 8 suggest that the initially heterolytically dissociated hydroxyl/hydride pair tends to evolve into two hydroxyls. They computed the homolytic H₂ dissociation in the presence of adjacent hydroxyls. The overall result is similar to two heterolytically dissociated H₂ (two hydroxyls and two hydrides). The DFT results do not seem to support the claim of excess hydride formation.*

Response: We appreciate the reviewer's question. We have supplemented the computational experiments after considering the surface proportion of H species. Firstly, H₂ homolytic dissociation on the hydrogenated α -Ga₂O₃ (001) surface with the Ga-H/OH ratio of 1:1, shows a negative dissociation energy of -0.46 eV (**Figure 3d**). Furthermore,

it's also found that the H₂ homolytic dissociation on the hydrogenated α-Ga₂O₃ (001) surface with one OH, where the Ga-H/OH ratio is 2:1, was also thermodynamically preferred with the H₂ adsorption energy of -0.55 eV (**Figure R10, Figure S20 in Supporting information**), consistent with experimental results. Under different coverage of -OH, homolytic H₂ dissociation is thermodynamically favoured. It proves the presence of excess hydride formation on α-Ga₂O₃ surface and homolytic H₂ dissociation. Additionally, the modification has been added in revised **Manuscript** and **Supporting information**.

Page 8 in Manuscript:

‘Interestingly, we found that H₂ homolytic dissociation in the presence of neighboring hydroxyl groups with the final Ga-H/OH ratio of 1:1 shows a negative dissociation energy of -0.46 eV (Figure 3d). Further increasing the Ga-H/OH ratio to 2:1 also show the negative dissociation energy (-0.55 eV, Figure S20).’

Figure R10 (Figure S20 in Supporting information). H₂ adsorption energy by homolytic dissociation over hydrogenated and O-defective α-Ga₂O₃ (001). Red ball: O atom, grey ball: Ga atom, white ball: H atom.

Reviewer #2:

*Comment: The manuscript deals with the role of **heterolytic splitting** in hydrogenations on oxides. While most of the results rely on the computational simulations these are kept to the minimum and the real important data are neither presented nor discussed. I find that the manuscript is below the quality required for Nature Comm. and therefore I suggest submission to a more specific journal.*

Response: We greatly appreciate the pertinent comments from the reviewer regarding our work. In this work, we reported an unusual phenomenon of **H₂ homolytic dissociation (not heterolytic dissociation) on oxides** that researchers had rarely reported previously. **H₂ homolytic dissociation** is generally believed to occur on metals, whereas H₂ heterolytic dissociation is considered to occur on oxides. This work presents an innovatively pioneering work on **H₂ homolytic dissociation** on oxides, which revolutionizes the traditional understanding about H₂ dissociation mechanism and gains positive feedback from other reviewers. This discovery has solved the long-standing problem that has puzzled researchers so far, extended the application of oxides in hydrogenation reactions, and greatly increased the theoretical upper limit for the activation of H₂ on oxides.

In this manuscript, we developed an experimental method for *in situ* differentiation and quantification of hydrogen species on oxide surfaces for the first time (**TKA-IR and TKA-MS**). We employed *in situ fast time-resolved XANES*, *in situ XPS*, and *in situ time-resolved infrared spectroscopy* combined with transient kinetic analysis (TKA) to obtain the quantity and evolution process of hydrogen species on oxide surfaces, so as to demonstrate the pathway of **H₂ homolytic dissociation**. A large amount of spectral evidence proved the rationality of **H₂ homolytic dissociation on oxides**.

In order to verify our experimental conclusions and seek for the nature of **H₂ homolytic dissociation** on oxides, the computational simulations were conducted. We designed dozens of Ga₂O₃ models based on experimental characterization results, calculated thermodynamic stable surfaces, and calculated data such as barriers for oxygen vacancy generation, H-migration barriers, adsorption energies of H₂ dissociation on oxide surfaces. All the calculation results can confirm our experimental structure and H₂ homolytic

dissociation pathway on oxides.

We tried our best to respond to the valuable suggestions by the reviewer and enhance the rigour and scientific level of our work. We have implemented lots of additional computational simulations and experiments, and thoroughly revised our manuscript. We conducted theoretical simulation and calculation based on the results of a large number of *in-situ* experimental characterizations, and ensured that the calculation results and experimental results mutually support each other. Previously, the reviewer might misunderstand our central idea regarding **H₂ homolytic dissociation on oxides**. We believe that our revisions adequately address all the concerns raised by the reviewer and reached the quality required for *Nat. Commun.*.

Comment 1): *I would expect a full characterization of the three material phases. This shall include at least their corresponding structures, low lying surfaces, Bader charges and Density of States for all three phases and the careful comparison to literature and experimental values.*

Response: We thank the reviewer for the helpful advice. We have performed a detailed theoretical characterization of three Ga₂O₃ crystalline phases, including **bulk phase/surface coordination, the exposed surfaces, and density of states for these three phases**. The information on **crystal form, Bader charges, density of states, etc.** were also supplemented, which is consistent with the literature and our experimental studies including XANES, XPS and IR.

(1) Structures

As shown in **Figure R11 (Figure S13 in Supporting information)**, the α -Ga₂O₃ occupies the *R-3c* space group with all Ga atoms in octahedral coordination, the ϵ -Ga₂O₃ with the *Pna2₁* space group has a proportion of 75% octahedral sites and 25% tetrahedral sites, and the β -Ga₂O₃ crystallizes in the monoclinic *C2/m* space group with 50% octahedral and 50% tetrahedral sites. These results are consistent with the literature (*Physical Review B*, **2016**,

93, 115204, *Chemistry of Materials*, 2020, 32(19), 8460.) and the results of XANES about the Ga^{3+} coordination (**Figure 1**).

Figure R11 (Figure S13 in Supporting information). The models of bulk $\alpha\text{-Ga}_2\text{O}_3$, bulk $\epsilon\text{-Ga}_2\text{O}_3$ and bulk $\beta\text{-Ga}_2\text{O}_3$.

(2) Bader charges and Density of States

The Bader charge analysis shows that the valence state of Ga atom at octahedron site is around +1.7, slightly higher than that at tetrahedron site (+1.6) (**Figure R12, DOS**). The presence of Ga_{tet} site decreases the energy band gap, causing that the energy gap of $\alpha\text{-Ga}_2\text{O}_3$ is obviously higher than that of $\epsilon\text{-Ga}_2\text{O}_3$ and $\beta\text{-Ga}_2\text{O}_3$, as shown in **Figure R12**.

Figure R12. The total density of states of three Ga_2O_3 phases.

(3) Low lying surfaces

After the surface energy calculation, the identified most stable surfaces for α -Ga₂O₃, ε -Ga₂O₃ and β -Ga₂O₃ crystals are (001), (011) and (100) surfaces, respectively, as illustrated in Figure R13-R14 (Figure S12-S13 in Supporting information) and Table R1 (Table S3 in Supporting information), which are well consistent with literatures (*Physical Review B*, 2016, 93, 115204; *Chemistry of Materials*, 2020, 32(19), 8460-8470.).

Figure R13 (Figure S14 in Supporting information). The models of α -Ga₂O₃(001) surface, ε -Ga₂O₃(011) surface and β -Ga₂O₃(100) surface.

Table R1 (Table S3 in Supporting information). The surface energy and Wulff area ratio of three Ga₂O₃ crystals.

No.	Crystal	Surface	Surface energy (J/m ²)	Wulff Area ratio (%)
1	α -Ga ₂ O ₃	s_001	1.11	37
2	α -Ga ₂ O ₃	s_110	1.43	38
3	α -Ga ₂ O ₃	s_101	1.46	0
4	α -Ga ₂ O ₃	s_010	1.53	8
5	α -Ga ₂ O ₃	s_100	1.53	0

6	α -Ga ₂ O ₃	s_011	1.61	17
7	α -Ga ₂ O ₃	s_111	1.68	0
8	β -Ga ₂ O ₃	s_100	0.48	55
9	β -Ga ₂ O ₃	s_001	1.15	14
10	β -Ga ₂ O ₃	s_101	1.28	4
11	β -Ga ₂ O ₃	s_111	1.29	27
12	β -Ga ₂ O ₃	s_011	1.45	0
13	β -Ga ₂ O ₃	s_010	1.50	0
14	β -Ga ₂ O ₃	s_110	1.68	0
15	ϵ -Ga ₂ O ₃	s_011	0.86	66
16	ϵ -Ga ₂ O ₃	s_001	1.23	1
17	ϵ -Ga ₂ O ₃	s_101	1.31	19
18	ϵ -Ga ₂ O ₃	s_110	1.38	11
19	ϵ -Ga ₂ O ₃	s_111	1.39	3
20	ϵ -Ga ₂ O ₃	s_100	1.47	0
21	ϵ -Ga ₂ O ₃	s_010	1.49	0

α -Ga₂O₃

ϵ -Ga₂O₃

β -Ga₂O₃

Figure R14 (Figure S12 in Supporting information). The thermodynamic Wulff morphology of three Ga₂O₃ crystals.

Among, the α -Ga₂O₃ (001) and ϵ -Ga₂O₃ (011) surface can generate the surface O_v with the assistant of H₂ but the β -Ga₂O₃ (100) surface is hardly to be reduced by H₂ (**Figure R15-16 and Table R2**), well consistent with our H₂-TPR results (**Figure S11**). All these data has been added into the revised **Manuscript and Supporting Information**, which are also copied below.

Figure R15. The models and distance of Ga-Ga site on α -Ga₂O₃(001) surface with oxygen vacancy (O_v), ϵ -Ga₂O₃(011) surface with oxygen vacancy (O_v) and β -Ga₂O₃(100) surface.

Table R2. The oxygen vacancy formation energies of the most stable surfaces for three Ga₂O₃ crystals.

Surface	O _v coverage (ML)	E _{Ov} (eV)
α -Ga ₂ O ₃ (001)	0 -> 1/12	3.09
	1/12 -> 1/6	3.24
β -Ga ₂ O ₃ (100)	0 -> 1/8	4.22
ϵ -Ga ₂ O ₃ (011)	0 -> 1/8	2.49
	1/8 -> 1/4	3.09

Figure R16. Thermodynamic phase diagram for α -Ga₂O₃ (001), ϵ -Ga₂O₃ (011) and β -Ga₂O₃ (100) contacting with H₂ at different temperatures and H₂ partial pressures (P_{H_2}), the phase diagram is computed based on Gibbs free energy data for the reaction ($\text{Ga}_2\text{O}_3 + x\text{H}_2 \rightarrow \text{Ga}_2\text{O}_{3-x} + x\text{H}_2\text{O}$) from DFT calculations assuming a H₂O pressure of 0.1 kPa. P^0 represents the standard pressure (101.325 kPa).

Comment 2): Once hydroxylated the surface might change, again I would expect a full characterization.

Response: We thank the reviewer for the thoughtful comment. We have performed a Bader charge and density of states analysis of the hydrogenated O-defective α -Ga₂O₃ (001) and ϵ -Ga₂O₃ (011) surfaces. As illustrated in **Table R3 (Table S8 in Supporting information)** and **Figure R17-R19 (Figure S22, S23 in Supporting information)**, the presence of O_V leads to the reduction of Ga atoms to +1 valence state and the large electron occupation of the valence band maximum on Ga 4p orbital. After the formation Ga-H species, a covalent bond between Ga 4p and H 1s orbital is established and the electrons transfer from the high-energy-level Ga 4p orbital to the low-energy-level 4p-1s hybrid orbital, leading the reoxidation of Ga atom to around +1.4. The density of states analysis is consistent with the results of experiments including *in situ* XPS, IR and XANES. All these data presented below have been added into the revised **Manuscript** and **Supporting**

Information.

Page 9 in Manuscript:

‘Homolytic dissociation of H_2 is also thermodynamically favored on O-defective $\epsilon\text{-Ga}_2\text{O}_3(011)$ surface with the H_2 dissociation energy of -0.55 eV and the charge transfer from Ga to H (Figure S22-S23 and Table S8), whereas the $\beta\text{-Ga}_2\text{O}_3(100)$ surface cannot dissociate the H_2 .’

(1) Density of States after hydroxylation

Figure R17 (Figure S22a in Supporting information). The density of states of Ga 4p orbital for the O-defective $\alpha\text{-Ga}_2\text{O}_3(001)$ surfaces with/without the formation of GaH species.

Figure R18 (**Figure S22b in Supporting information**). The density of states of Ga 4p orbital for the O-defective ϵ -Ga₂O₃ (011) surfaces with/without the formation of GaH species.

Figure R19 (**Figure S23 in Supporting information**). The models of OH and Ga-H site on α -Ga₂O₃ (001) surface with oxygen vacancy (O_v), ϵ -Ga₂O₃ (011) surface with oxygen vacancy (O_v) and β -Ga₂O₃ (100) surface.

Table R3 (**Table S8 in Supporting information**). The valence state analysis of different Ga atoms for three Ga₂O₃ crystals based on Bader charge.

Crystal	Environment	Ga coordination	Ga valence state (e ⁻)
α -Ga ₂ O ₃	bulk	Ga _{6c[oct]}	+1.7
	(001)	Ga _{6c}	+1.6
	(001)-O _v	Ga _{2c}	+1.0
	(001)-O _v -H	Ga _{2c}	+1.3
ε -Ga ₂ O ₃	Bulk	Ga _{6c[oct]}	+1.7
	Bulk	Ga _{4c[tet]}	+1.7
	(011)	Ga _{4c}	+1.6
	(011)-O _v	Ga _{3c}	+1.0
	(011)-O _v -H	Ga _{3c}	+1.4
β -Ga ₂ O ₃	Bulk	Ga _{6c[oct]}	+1.8
	Bulk	Ga _{4c[tet]}	+1.7
	(100)	Ga _{5c}	+1.7

(2) Surface species and structure after hydroxylation

A large amount of spectral and energy spectral experimental evidence also suggested that the surface structure, especially electronic structure, has indeed changed after hydroxylation. **These results are the basis for our theoretical calculation modelling and supported the calculation results.**

Firstly, *in-situ* Ga 2p XPS spectrum of the hydroxylated sample shows the appearance of a peak at a higher binding energy position, confirming the reoxidation process of Ga (**Figure S46**). The infrared characteristic peak of Ga-H also undergoes a red shift as the degree of hydroxylation deepens, indicating changes in Ga-H bond energy and electronic density of

Ga³⁺ (**Figure S23**). These experimental data and computational simulation results complement each other.

Figure S46 *In-situ* XPS spectra of Ga 3d for Ga₂O₃ samples. The blue, red and green colors refer to treatment conditions of H₂, CO₂ and H₂+CO₂ (H₂/CO₂= 3 (v/v)), respectively. Reaction conditions: 350 °C, 3 MPa.

Figure S23 Evolution of Ga-H band position of Ga₂O₃ samples during H₂ treatment. The Ga-H peaks were normalized in intensity for better comparison.

On the other hand, the surface structure and surface species of gallium oxide did change after hydroxylation. The results of *in situ* ultrafast time-resolved XANES showed that after hydroxylation, except for a decrease in the height of white line on α -Ga₂O₃, the positions and shapes of white line peak remained basically unchanged during reduction, indicating that the reduction process produced more oxygen vacancies on α -Ga₂O₃, but had little effect on the coordination environment of Ga³⁺ for the different Ga₂O₃ phase (**Figure S4**). In addition, *in situ* infrared spectroscopy during hydrogen-treatment was also conducted.

It is found that there were no additional surface species generated except for hydroxyl groups and Ga-H on the Ga_2O_3 surface. Thus, the hydroxylation process only involves changes in hydroxyl groups spectrum and Ga-H (Figure R20).

Figure S4 Ga K-edge XANES of fresh Ga_2O_3 and Ga_2O_3 samples contacting with H_2 for 15 min at $350\text{ }^\circ\text{C}$, 1 MPa.

Figure R20. *In-situ* FTIR of $\alpha\text{-Ga}_2\text{O}_3$ samples contacting Ar to H_2 at $350\text{ }^\circ\text{C}$.

Comment 3): The TS characterization needs to include the transition states vibrational characterization.

Response: We are grateful for the valuable advice from the reviewer. All transition states

are verified by frequency analysis which shows only one imaginary frequency along the reaction direction. We provide the structure data in **Supporting data** and the imaginary frequencies of key transition states are also shown below (**Table R4, Table S10 in Supporting information**).

Table R4 (Table S10 in Supporting information). The transition states analysis.

Initial State	Final State	TS imaginary frequency (cm ⁻¹)
CO ₂ *+4H*	HCOO*+3H*	191.0
HCOO*+5H*	H ₂ COOH*+3H*	304.7
H ₂ COOH*+3H*	H ₂ CO*+H ₂ O*+2H*	846.8
CH ₂ O*+2H*	CH ₃ O*+H*	1241
CH ₃ O*+H*	CH ₃ OH*	1120.9
CO ₂ *+2H*	COOH*+H*	101.9
COOH*+H*	OH*+H*+CO	280.1
OH*+H*	H ₂ O	89.6

Comment 4): All the structures need to be uploaded to a suitable database for other researchers to reproduce the data found here.

Response: We thank the reviewer for the suggestion. We provide all the structures in the **Supporting Data**.

Besides, the detailed methods about DFT calculation are also added in the revised **Supporting information**.

Page S8 in Supporting information

DFT calculation. All DFT calculations are performed by using the plane wave VASP code,³ where electron-ion interaction is represented by the projector augmented wave pseudopotential.^{4,5} The exchange functional utilized is the spin-polarized GGA-PBE.⁶ The kinetic energy cutoff is set as 450 eV. The first Brillion zone k -point sampling utilizes the Monkhorst-Pack scheme with an automated mesh determined by 18 times of the reciprocal lattice vectors. The energy and force criterion for convergence of the electron density and structure optimization are set at 10^{-6} eV and 0.05 eV/Å, respectively.

Ab-initio thermodynamics analyses. To determine the equilibrium O_v concentration in Ga_2O_3 surfaces, the ab-initio thermodynamics analyses have been performed where the formula

are used to compute the free energy change as a function of temperature and H_2 partial pressure. To determine the Gibbs free energy change (ΔG) per formula unit (f. u.) for the above reactions, one needs to compute

$$\Delta G(p, T) = G[GaO_x](p, T) + (y - x) * \mu[H_2O](p, T) - G[GaO_y](p, T) - (y - x) * \mu[H_2](p, T) \quad (16)$$

where G is the Gibbs free energy of surfaces and μ is the chemical potential for molecules. The $G[X]$ can be approximated by their DFT total energy $E[X]$ with appropriate inclusion of zero-point-energy (ZPE), since it is known that the vibration entropy and the pV term contributions of solid phases are negligibly small. The chemical potential for molecules $\mu[X]$ can be calculated as follows:

$$\mu[X](p, T) = E[X] + ZPE[X] + [H[X](p^0, T) - H[X](p^0, 0K) - TS[X](p^0, T) + k_B T \ln \frac{p}{p^0}] \quad (17)$$

where enthalpy (H) and entropy (S) terms are taken from the standard thermodynamics data.

Comment 5): As points 1-3 are crucial to understand why the different systems behave so differently I cannot recommend the work for publication.

Response: We appreciate the thought-provoking comment provided by the reviewer. According to the suggestions of the reviewers, we have provided detailed structural models, barde charge and density of states for different Ga₂O₃, taking into account various exposed crystal planes. Comment 1-3 has been responded point by point. The calculations can be beard out by corresponding experimental evidence, which is well founded. All structural parameters are consistent with the data in the literature. We hope that the above experiments and improvements can answer the reviewers' questions and sincerely hope that our revised manuscript could be reconsidered.

It is worth emphasizing again that our research offers a quantitative and time-resolved analysis of hydrogen activation on metal oxides, specifically shedding light on the long-standing puzzle of **homolytic dissociation as opposed to the heterolytic pathway**. The concurrency of homolytic and heterolytic dissociation on oxide contrasted the general view of homolytic dissociation on metals and heterolytic dissociation on oxides, enabling oxides with “metal-like” hydrogenation reactivity.

Hydrogen activation on solid surfaces holds significant importance in catalytic processes involving hydrogen evolution, as well as for fundamental mechanistic studies to develop catalyst design principles. Metal oxides play a crucial role in hydrogen-related catalysis and traditionally believed to catalyze hydrogen dissociation heterolytically. Limits associated with heterolytic dissociation, i.e., low surface coverage and weak activity of hydride, lead to flaws in explaining the high reactivity of oxides in some hydrogen-related catalysis (e.g., CO₂ and CO hydrogenation).

By developing the innovative transient kinetic analysis infrared spectroscopy (TKA-IR) and mass spectroscopy (TKA-MS), we have successfully quantified surface hydrides (H*) and hydroxyls (OH*) and tracked their evolution on Ga₂O₃ during hydrogen dissociation and catalytic hydrogenation. We have identified different formation rates for H* and OH*

and observed a high H/OH ratio of 5.6, deviating greatly from the expected stoichiometric value of 1 in heterolytic dissociation. Consequently, we are able to provide a quantitative distinction of homolytic dissociation of hydrogen from the heterolytic pathway on Ga₂O₃. Homolytic dissociation leads to a high hydride-to-surface Ga ratio of 1.6 and a hydrogen dissociation activity that is 1-order-of-magnitude higher than that achieved through heterolytic dissociation. These high coverage hydrides play a crucial role in driving the conversion of inert CO₂ and intermediates such as HCOO*.

We firmly believe that our work not only provides a clarification of the long-standing hydrogen dissociation pathway on oxides, but also offers a guidance to enhance the hydrogenation ability of oxides by promoting homolytic dissociation. Given the interdisciplinary nature of our study, we are confident that it will attract a broad readership in the field of chemistry, catalysis, surface science and materials science. We believe that the Nature Communications is the suitable platform for the publication of our paper.

Reviewer #3:

Comment: *This is an very interesting work with detailed spectroscopy evidence for the presence of two H_2 dissociation channels on Ga_2O_3 and how they are impacted by the phase/surface structure of Ga_2O_3 . The finding was also used to explain the large catalytic differences observed in several hydrogenation reactions over different Ga_2O_3 phases. I commend the authors for a very careful investigation on the fundamentals of H_2 interaction and reaction over an oxide surface. The work can be accepted for publication after some revisions as noted below.*

Response: We appreciate the reviewer for his/her carefully reading of the manuscript and the professional comments. In the revised version, we have made all the necessary additions of experiments and discussions (highlighted with yellow background in the revised manuscript and supporting information) to better explain the conclusion of this work. Our point-by-point responses are listed below.

Comment 1): *1) is the intrinsic activity (i.e., TOF) of the hydrides on octahedra Ga^{3+} the same for the three different phases? or is it depend on the Ga_2O_3 phase? The IR switch experiment results could be used to derive the intrinsic rates of the surface hydrides on alpha and beta- Ga_2O_3 .*

Response: We appreciate the helpful suggestion from the reviewer. The intrinsic activity of the hydrides on different phases is a very important part for this work. We calculated the conversion rate of Ga-H reacted with CO_2 via the IR switch experiment (**Figure R21**). The GHSV is as high as 120000 ml/g_{cat}/h, where the diffusion is eliminated and the intrinsic rates of the surface hydrides would be acquired. We found that the intensity of Ga-H peak declined with the CO_2 inlet into the transmission cell. Meanwhile, the $HCOO^*$ and CH_3O^* species appeared and were increasing with CO_2 inlet. Differential processing was performed on the amounts of Ga-H species as a function of CO_2 exposure time to obtain the conversion rate of hydride in three different phases. It is found conversion rate of the hydrides on octahedra Ga^{3+} of *alpha*- Ga_2O_3 is two times higher than hydrides on octahedra Ga^{3+} of the ϵ phase and is much higher than tetraordinated Ga^{3+} of *beta*-

Ga_2O_3 (Figure R22). Thus, we speculated that the intrinsic activity of the hydrides on octahedra Ga^{3+} did depend on the Ga_2O_3 phase.

Figure R21. *In-situ* FTIR spectra of surface species when switching H_2 to CO_2 over Ga_2O_3 at 350°C .

Figure R22. **a** The amount of surface Ga-H contacting CO_2 at 350°C over different Ga_2O_3 phase. **b** Conversion rate of Ga-H reacted with CO_2 *in-situ* FTIR spectra.

Comment 2): it is interesting to compare the reactivity of the surface hydrides on Ga_2O_3 in the hydrogenation reactions with Ga_2O_3 supported metal catalysts (such as Cu for CO_2 to methanol or CO). This will provide an idea of how reactive these hydrides are in comparison to hydrogenation assisted with a metal surface.

Response:

We appreciate the helpful suggestion of reviewers. Using Ga_2O_3 with different phase as

support, Cu/Ga₂O₃ catalysts were prepared by equal volume impregnation method with ca. 5wt% loading amount of Cu. The catalytic test for CO₂ hydrogenation to methanol and CO is conducted at 250-500°C and 4 MPa. The test at high GHSV (24000 ml/g_{cat}/h) is also carried out to obtain the intrinsic activity of Cu/Ga₂O₃. Methanol is produced mainly under low-temperature condition (250-350°C) and CO is produced under high-temperature condition (350-500°C) (**Figure R23**). The methanol yield of Cu/α-Ga₂O₃ reached maximum of 0.4 mmol/g_{cat}/h at 300 °C, which is twice higher than that of Cu/β-Ga₂O₃. Methanol synthesis is believed to be promoted by the stronger H₂ activation of Cu/α-Ga₂O₃ (**Figure R24a**).

Cu/α-Ga₂O₃ produces more methane at high temperature of 500°C, and the CO yield of Cu/α-Ga₂O₃ is slightly higher in the range of 350 to 450°C than Cu/β-Ga₂O₃ (**Figure R24b**). Because the reaction order of H₂ in CO synthesis (RWGS) is close to 0, the generation of CO is less affected by the H₂ activation ability of catalyst (*J. Am. Chem. Soc.* **2020**, 142, 19523-19531). Thus, the promotion effect of Cu/α-Ga₂O₃ for RWGS is not as obvious as that for methanol synthesis.

Conclusively, the above results also indicate the stronger hydrogenation ability of Cu/α-Ga₂O₃.

Figure R23. Reaction performance of CO_2 hydrogenation over $\text{Cu}/\text{Ga}_2\text{O}_3$. Reaction conditions: $\text{H}_2/\text{CO}_2 = 3$ (v/v), 250-500 °C, 4 MPa, 3600 ml/g_{cat}/h (a, c), 24000 ml/g_{cat}/h (b, d).

Figure R24. a Yield of methanol in CO_2 hydrogenation over $\text{Cu}/\text{Ga}_2\text{O}_3$. b Yield of CO in CO_2 hydrogenation over $\text{Cu}/\text{Ga}_2\text{O}_3$. Reaction conditions: $\text{H}_2/\text{CO}_2 = 3$ (v/v), 250-500 °C,

4 MPa, 24000 ml/g_{cat}/h.

We also conducted TKA-MS experiments to quantify surface hydrogen species over Cu/Ga₂O₃ (**Figure R25**). Using inert SiO₂ loaded with the same amount of Cu as a reference catalyst, we found that there are relatively fewer metal-hydrogen bonds on Cu (**Figure R26**). After deducting Cu-H, the Cu/ α -Ga₂O₃ was measured to have a higher density of Ga-H than other samples, which is ten times higher than -OH (Ga-H/Ga-OH= 11.0 for Cu/ α -Ga₂O₃). Thus, the introduction of Cu further increased the amount of Ga-H over Ga₂O₃ by H₂ spillover. Above all, homolytic H₂ dissociation also occurs on Cu/ α -Ga₂O₃ to produce high-density of Ga-H, which is similar to the α -Ga₂O₃ (**Figure R27**).

Figure R25. **a, c** Hydrogen consumption with $m/z=2$ (H₂, red), $m/z=3$ (HD, blue) of Cu/Ga₂O₃ during H₂-exchange experiment at 150 °C after the catalysts were saturated with D₂ at 350 °C. **b, d** Hydrogen consumption with $m/z=2$ (H₂, red), $m/z=3$ (HD, blue) of

Cu/Ga₂O₃ during H₂-exchange experiment at 350 °C after the catalysts were saturated with D₂ at 350 °C.

Figure R26. Hydrogen consumption with $m/z=2$ (H₂, red), $m/z=3$ (HD, blue) of Cu/SiO₂ during H₂-exchange experiment at 350 °C after the catalysts were saturated with D₂ at 350 °C.

Figure R27. **a** Amount of surface hydrogen species over Ga₂O₃. **b** Amount of surface hydrogen species over Cu/Ga₂O₃.

Similarly, when hydrogen is introduced into the *in-situ* IR transmission cell, Ga-H can be observed to appear on the surface of α-Ga₂O₃ (**Figure R28**). When the H₂ is switched to CO₂, surface Ga-H reacted with CO₂ and was converted into intermediates such as formate and methoxy (**Figure R29**). Therefore, the high-density Ga-H generated by the

introduction of Cu can promote the CO₂ hydrogenation reaction and improve both the CO₂ conversion rate and the yield of methanol, compared with Ga₂O₃.

Figure R28. *In-situ* FTIR spectra of surface species over Cu/Ga₂O₃ when contacting H₂ at 350°C.

Figure R29. *In-situ* FTIR spectra of surface species when switching H₂ to CO₂ over Cu/ α -Ga₂O₃ at 350°C.

Comment 3): a scheme should be provided on how the 2nd dissociation channel, the homolytic dissociation, occurs and the migration of H to form extra hydride.

Response: We thank the reviewer for the advice. We have provided a scheme on how the 2nd dissociation channel, the homolytic dissociation, occurs and the migration of H in the revised manuscript (**Figure S30 (Figure S21 in Supporting information)**).

As illustrated in **Figure R30 (Figure S21 in Supporting information)**, the oxygen vacancies were generated under H₂ flow initially. Subsequently, the H₂ heterolytic dissociation occurred on the vacancies to form (Ga)H⁻ hydride and (O)H⁺ proton. The resulting hydride tends to migrate to the neighbouring O atom, forming hydroxyl group and releasing energy of 0.58 eV. We found that H₂ homolytic dissociation in the presence of neighbouring hydroxyl groups showed a negative dissociation energy of -0.46-0.55 eV (**Figure S20**). On the contrary, H₂ homolytic dissociation without the prior H₂ heterolytic dissociation shows the positive dissociation energy (+0.28 eV). It suggests that H₂ heterolytic dissociation is the prerequisite for homolytic dissociation (**Figure S21**).

Figure R30 (Figure S21 in Supporting information). Scheme of the homolytic dissociation and the migration of H over $\alpha\text{-Ga}_2\text{O}_3$.

Comment 4): some of the claims/sentences need be clarified. for examples, on P3 Line 34-35: "breaking the ceiling of hydrogenation capacity for oxides", what is the ceiling of hydrogenation capacity of oxides? I think the authors meant the capacity of surface hydrogen is above the ceiling of a monolayer H on an oxide surface? on α -Ga₂O₃, the authors claimed a 1.6 ratio for H/surface Ga³⁺, what is the ratio of H/surface O? is the total H (hydride + proton (OH)) density higher than the total surface atom density (Ga+O)? Anyway, the sentence needs revision and more relevant discussion is warranted in later section.

Response: We thank the reviewer for pointing out the confusing expression. We are sorry for not considering the H species coverage on O. We supplemented the ratio of H to surface O and the ratio of total H (hydride + proton (OH)) to the total surface atom density (Ga+O) (**Table R5 (Table S7 in Supporting information)**). The expression has been modified in the revised manuscript (highlighted with yellow background). The relevant discussion has been also supplemented in the revised manuscript.

Page3 Line 34-35 in Manuscript:

"improving the hydrogenation ability for oxides"

The ratio of H/surface O and total H density are listed as following (**Figure R5, Table S7 in Supporting information**). The density of surface O is calculated based on Ga/O from the XPS data. The ratio of H/surface O for α -Ga₂O₃, ϵ -Ga₂O₃ and β -Ga₂O₃ are 0.24, 0.38 and 0.08, respectively. And, the total H density is lower than the total surface atom density (Ga+O). The coverage of H* on O or total surface atom is not saturated. Thus, we modified the expression as "Homolytic dissociation of H₂ on oxides improved the hydrogenation capacity of oxides." All these data are added into the **Supporting Information** and are also presented following.

Table R5 (Table S7 in Supporting information). Coverage of activated hydrogen species

on Ga₂O₃ surface.

Catalysts	H _[surface] /Ga _[surface] (mol/mol)	OH _[surface] /Ga _[surface] (mol/mol)	H _[surface] /O _[surface] (mol/mol)	Total H (H+OH) density (mmol/g)	Total surface atom (Ga+O) density (mmol/g)
α-Ga ₂ O ₃	1.60	0.29	0.24	1.77	2.05
ε-Ga ₂ O ₃	0.67	0.51	0.38	1.27	2.54
β-Ga ₂ O ₃	0.08	0.11	0.08	0.10	1.30

*Relevant discussion is warranted in **Page10 in Manuscript**:*

'It is worth noting that the coverage of H on metal ions was calculated close to saturation. When considering the coverage of H* on surface O of oxide (**Table S7**), there still allows potential in promoting the adsorption capacity of H*.'*

Summary of revision (Manuscript Number: NCOMMS-23-33063-T)	
Comment	Brief Revision
Reviewer #1:	
The quantification of proton and hydride, which was based on the IR assignments of GaO-H/GaO-D and Ga-H/Ga-D bands and the MS quantification of H₂ consumption (or HD formation? This does not seem to be clear).	The quantification of proton and hydride was based on the MS quantification of H₂ consumption . We have bolded and marked the right vertical axis of Figure 2c and Figure 2e in red to highlight this point and make it clear. Additionally, more detailed description of the quantitative process was added in Supporting information .
The authors should provide full spectra of their IR results in order to give readers a complete picture.	We have provided the full spectra for all of the IR results, including H ₂ -reduction, TKA-IR and H ₂ -D ₂ exchange experiment in the revised manuscript (Figure R1-R3).
The feature (negative peak on the left, positive band on the right) is likely associated with the redshift of an existing IR band of Ga₂O₃ by switching atmosphere from Ar to D₂.	The negative peak may be attributed to the spectral fluctuation caused by background perturbation. According to Reviewer #1's suggestion, we conducted sufficient degassing and drying on the sample, after which the infrared spectrum with the blank transmission cell as background became flat, and the negative peak became weak (Figure R3). The new IR spectra also demonstrated the H ₂ homolytic dissociation on α -Ga ₂ O ₃ .
Without seeing the full spectra, it is hard to justify their band assignments. On the other hand, is it possible to measure the IR spectra with KBr or other background to show the actual IR features?	The full IR spectra, acquired and reported with the blank transmission cell as background, have been provided in the revised files (Figure 2b and 2d, Figure R1-R3).
Do they have a more explicit way to quantify the amount of newly formed proton and hydride upon exposure in H₂? How about NMR?	Quasi in-situ ¹ H-NMR was conducted. The amount of Ga-H is much higher than Ga-OH over α -Ga ₂ O ₃ , while Ga-H/-OH of β -Ga ₂ O ₃ is 0.5, also suggesting the homolytic dissociation of H ₂ on α -Ga ₂ O ₃ (Figure R4-R5).
Fig S13 (b, d, e) shows an increasing H₂ signal but the HD signal first increases and then drops. It seems that H₂ was consumed without forming HD at the beginning. What caused the delay in HD formation?	The quick adsorption of H ₂ , the slow desorption of HD on α -Ga ₂ O ₃ and the possible mechanism of H-D bond formation followed by Ga-D bond breaking in H ₂ -D ₂ exchange reaction may root in the delay in HD formation, proved by TKA, H ₂ -TPD and isothermal chemisorption

	experiments (Figure R6-R9).
4. The DFT results discussed on page 8 suggest that the initially heterolytically dissociated hydroxyl/hydride pair tends to evolve into two hydroxyls. They computed the homolytic H₂ dissociation in the presence of adjacent hydroxyls. The overall result is similar to two heterolytically dissociated H₂ (two hydroxyls and two hydrides). The DFT results do not seem to support the claim of excess hydride formation.	We found that the H ₂ homolytic dissociation on the hydrogenated α-Ga ₂ O ₃ (001) surface with the Ga-H/OH ratio is 1:1 and 2:1, were both thermodynamically preferred with the H ₂ dissociation energy of -0.46 and -0.55 eV, respectively (Figure R10). It proves the presence of excess hydride formation (homolytic dissociation of H ₂) on α-Ga ₂ O ₃ surface.
Reviewer #2:	
I would expect a full characterization of the three material phases. This shall include at least their corresponding structures, low lying surfaces, Bader charges and Density of States for all three phases and the careful comparison to literature and experimental values	There is information on crystal form, bulk phase/surface coordination, etc. were supplemented in revised manuscript. We also have performed a detailed characterization of three Ga ₂ O ₃ crystalline phases, including the exposed surfaces, Bader charges and density of states (Figure R11-R16, Table R1-R2).
Once hydroxylated the surface might change, again I would expect a full characterization.	We have performed the Bader charge and density of states analysis of the hydrogenated O-defective Ga ₂ O ₃ surfaces. (Figure R17-R19, Table R3). A large amount of experimental evidence consisting of spectra and energy spectra also suggested that the surface structure, especially electronic structure, had changed after hydroxylation, which is consistent with the DFT calculation.
The TS characterization needs to include the transition states vibrational characterization.	All transition states are verified by frequency analysis which shows only one imaginary frequency along the reaction direction. We provide the structure data in Supporting data and the imaginary frequencies of key transition states are also shown below (Table R4).
All the structures need to be uploaded to a suitable database for other researchers to	We provide all the structures in the Supporting Data .

reproduce the data found here.	
Reviewer #3:	
Is the intrinsic activity (i.e., TOF) of the hydrides on octahedra Ga³⁺ the same for the three different phases? or is it depend on the Ga₂O₃ phase?	We calculated the conversion rate of Ga-H reacted with CO ₂ via the IR switch experiment (Figure R21). We found that the conversion rate of the hydrides on octahedra Ga ³⁺ of α-Ga ₂ O ₃ is two times higher than the ε phase and β phase (Figure R22). Thus, we concluded that the intrinsic activity of the hydrides on octahedra Ga ³⁺ depended on the Ga ₂ O ₃ phase.
it is interesting to compare the reactivity of the surface hydrides on Ga₂O₃ in the hydrogenation reactions with Ga₂O₃ supported metal catalysts (such as Cu for CO₂ to methanol or CO). This will provide an idea of how reactive these hydrides are in comparison to hydrogenation assisted with a metal surface.	Cu/α-Ga ₂ O ₃ was measured to have a higher density of Ga-H than other samples via TKA-MS experiments. The high-density Ga-H generated by the introduction of Cu can promote the CO ₂ hydrogenation reaction and improve both the CO ₂ conversion rate and the yield of methanol, compared with pure Ga ₂ O ₃ (Figure R23-S29).
a scheme should be provided on how the 2nd dissociation channel, the homolytic dissociation, occurs and the migration of H to form extra hydride.	We have provided a scheme on how the 2nd dissociation channel and the homolytic dissociation occur as well as the migration of H to form extra hydride in the revised manuscript (Figure R30).
some of the claims/sentences need be clarified. for examples, on P3 Line 34-35: "breaking the ceiling of hydrogenation capacity for oxides", what is the ceiling of hydrogenation capacity of oxides? I think the authors meant the capacity of surface hydrogen is above the ceiling of a monolayer H on an oxide surface? on α-Ga₂O₃, the authors claimed a 1.6 ratio for H/surface Ga³⁺, what is the ratio of H/surface O? is the total H (hydride + proton (OH)) density higher than the total surface atom density (Ga+O)? Anyway, the sentence	As the reviewer suggested, we have modified the expression in the revised manuscript (highlighted with yellow background). The ratio of H/surface O and the total H (hydride + proton (OH)) density are listed in Table R5 . The total H density is lower than the total surface atom density (Ga+O).

needs revision and more relevant discussion is warranted in later section.	
---	--

REVIEWERS' COMMENTS

Reviewer #1 (Remarks to the Author):

The authors have nicely addressed my concerns by conducting additional experiments and providing more detailed explanations. Therefore, I recommend acceptance without further revision.

Reviewer #2 (Remarks to the Author):

The authors have improved the manuscript significantly. I appreciate the effort in clarifying all the different aspects I raised in my previous comments and thus I can now recommend the work for publication.

Reviewer #3 (Remarks to the Author):

The authors did a great job addressing my previous comments. However, some of the new information can be useful for the paper and the readers, so I suggest the authors to incorporate some of the main results (such as reactivity difference of Ga-H on Ga₂O₃ with different phases, and the reactivity differences between Ga₂O₃ and Cu/Ga₂O₃) in the response letter including Figures R22-R29 into the main manuscript or the supporting information. Once these are addressed, the paper can be accepted.

Reviewer #1 (Remarks to the Author):

The authors have nicely addressed my concerns by conducting additional experiments and providing more detailed explanations. Therefore, I recommend acceptance without further revision.

Response: We appreciate the reviewer for his/her professional and positive comments.

Reviewer #2 (Remarks to the Author):

The authors have improved the manuscript significantly. I appreciate the effort in clarifying all the different aspects I raised in my previous comments and thus I can now recommend the work for publication.

Response: We appreciate the reviewer for his/her carefully reading and the professional suggestion.

Reviewer #3 (Remarks to the Author):

The authors did a great job addressing my previous comments. However, some of the new information can be useful for the paper and the readers, so I suggest the authors to incorporate some of the main results (such as reactivity difference of Ga-H on Ga₂O₃ with difference phases, and the reactivity differences between Ga₂O₃ and Cu/Ga₂O₃) in the response letter including Figures R22-R29 into the main manuscript or the supporting information. Once these are addressed, the paper can be accepted.

Response: We thank the reviewer for the professional comments and the valuable suggestion. As the reviewer suggested, we have made the additions of experiments and discussions (Figures R22-R29 in previous response file) into manuscript to show the reactivity difference of Ga-H on Ga₂O₃ with difference phases and the reactivity differences between Ga₂O₃ and Cu/Ga₂O₃.

Line 2-4 in Page 15 of the main manuscript:

Above all, the intrinsic activity of the hydrides to hydrogenate CO₂ on octahedra Ga³⁺ of α-Ga₂O₃ is higher than that tetrahedral Ga³⁺ of β-Ga₂O₃ (Supplementary Fig. 47).

Figure S47. **a** The amount of surface Ga-H contacting CO₂ versus time at 350°C over different Ga₂O₃ phase. **b** Conversion rate of Ga-H reacted with CO₂ versus time.

Line 14-21 in Page 15 of the main manuscript:

Additionally, we also quantified surface hydrogen species over Cu/Ga₂O₃ (Supplementary Fig. 50-52). Cu/α-Ga₂O₃ was measured to have a much higher density of Ga-H than α-Ga₂O₃ and Cu/β-Ga₂O₃, which showed high activity to convert CO₂ into formate and methoxy intermediates (Supplementary Fig. 53-54). As a result, the methanol yield of Cu/α-Ga₂O₃ reached maximum of 0.4 mmol/g_{cat}/h at 300 °C, which is twice higher than that of Cu/β-Ga₂O₃ (Supplementary Fig. 55-56). This result proved that the α-Ga₂O₃ can serve as the promising material synergizing with metal to activate H₂ and further promote the CO₂ hydrogenation reaction.

Figure S50. **a, c** Hydrogen consumption with $m/z=2$ (H_2 , red), $m/z=3$ (HD, blue) of Cu/Ga₂O₃ during H₂-exchange experiment at 150 °C after the catalysts were saturated with D₂ at 350 °C. **b, d** Hydrogen consumption with $m/z=2$ (H_2 , red), $m/z=3$ (HD, blue) of Cu/Ga₂O₃ during H₂-exchange experiment at 350 °C after the catalysts were saturated with D₂ at 350 °C.

Figure S51. Hydrogen consumption with $m/z=2$ (H₂, red), $m/z=3$ (HD, blue) of Cu/SiO₂ during H₂-exchange experiment at 350 °C after the catalysts were saturated with D₂ at 350 °C.

Figure S52. a Amount of surface hydrogen species over Ga₂O₃. **b** Amount of surface hydrogen species over Cu/Ga₂O₃.

Figure S53. *In-situ* FTIR spectra of surface species over Cu/Ga₂O₃ when contacting H₂ at 350°C.

Figure S54. *In-situ* FTIR spectra of surface species when switching H₂ to CO₂ over Cu/α-Ga₂O₃ at 350°C.

Figure S55. Catalytic performance of CO₂ hydrogenation over Cu/Ga₂O₃. Reaction conditions: H₂/CO₂= 3 (v/v), 250-500 °C, 4 MPa, 3600 ml/g_{cat}/h (a, c), 24000 ml/g_{cat}/h (b, d).

Figure S56. a Yield of methanol in CO₂ hydrogenation over Cu/Ga₂O₃. **b** Yield of CO in CO₂ hydrogenation over Cu/Ga₂O₃. Reaction conditions: H₂/CO₂= 3 (v/v), 250-500 °C, 4 MPa, 24000 ml/g_{cat}/h.